# ChaoticFuzz: Fuzzy-Based Graph Representation for Spatiotemporal Learning

## Abstract

Chaos theory addresses a unique class of dynamical systems that, despite operating under deterministic rules, exhibit extreme sensitivity to initial conditions, where even minor perturbations can lead to vastly different trajectories. Rather than modeling such systems directly in the time domain, chaotic time series are often reconstructed in higher-dimensional phase spaces, where latent structures such as attractors and fixed points emerge. However, forecasting within this context remains challenging, as conventional time series models are ill-suited to capture the complex interplay between spatial and temporal dynamics. In this work, we propose ChaoticFuzz, a novel framework that transforms univariate chaotic time series into graph-structured data by leveraging fuzzy clustering in phase space. Our framework encodes temporal trajectories and fuzzy membership degrees to construct weighted graphs, enabling the application of Graph Neural Networks for accurate long-term prediction. This graph-based representation preserves essential spatiotemporal patterns that are often lost in traditional approaches. Experiments on several benchmark chaotic systems demonstrate that ChaoticFuzz significantly outperforms state-of-the-art methods, which reflect a model's ability to follow complex dynamical behavior.

## 1 Introduction

Time Series (TS) analysis plays a fundamental role in understanding systems, enabling the prediction of future events, the diagnosis of current states, and the detection of behavioral shifts over time (Box et al., 2015). Unlike traditional data, which are typically assumed to be independent and identically distributed (i.i.d.), time series exhibit complex temporal dependencies that must be modeled to ensure accurate and reliable predictions. Traditional statistical (e.g., ARIMA) and machine learning approaches (e.g., RNNs) analyze TS by modeling their dependencies through additive or multiplicative components of seasonality, trend, and noise. However, when TS exhibit nonlinear interactions and chaotic behavior, where observations lie in an equilibrium state that is extremely sensitive to perturbations, these models often fail to capture underlying temporal dependencies.

To overcome this problem, one effective approach is to reconstruct their observations in a higher-dimensional space known as the phase space (Alligood et al., 1997). In this representation, even univariate time series can reveal complex structures such as fixed points, and strange attractors that are not easily observable in the time domain. By mapping the time series into phase space, the evolving trajectories of the system become more structured and stable, enabling a clearer analysis of its long-term dynamics.

This reconstruction in phase space reveals a promising opportunity for time series modeling: transforming the unfolded structure into a graph, where observations exhibiting similar behavior are grouped into the same node, and their temporal or dynamic relationships are represented as edges. This graph-based representation enables a new perspective for capturing the complex, nonlinear dependencies inherent in chaotic systems. Figure 1 illustrates the core components of the proposed ChaoticFuzz framework. The pipeline begins by unfolding the input time series into a phase space to reveal the underlying dynamics and geometric structures, such as attractors and trajectories. A fuzzy clustering algorithm is then applied in the reconstructed space to assign degrees of membership to each observation, resulting in soft partitions that reflect uncertainty and overlap among clusters. These clusters serve as the nodes of a graph, while membership-based similarities define the edge

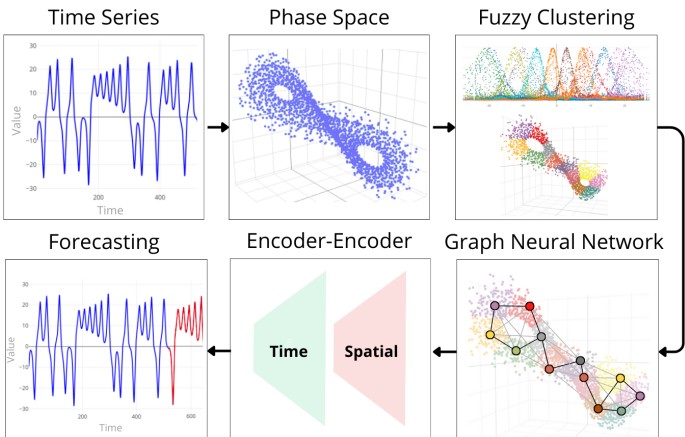

Figure 1: The ChaoticFuzz framework. The process begins unfolding the time series into a phase space to capture the system's underlying dynamics. Fuzzy clustering is applied to identify soft clusters and assign degrees of membership, leading to the construction of a graph structure. This graph is processed by GNN, enhanced with a global attention mechanism, to perform accurate forecasting of chaotic time series.

weights. The resulting graph structure encodes both temporal and spatial relationships and is subsequently processed by a Graph Neural Network (GNN). To enhance the model's ability to capture long-range dependencies and global patterns, we incorporate a global attention mechanism (Liu et al., 2021) prior to generating the final forecast. This step is referred to in our manuscript as the phase-space encoder-encoder, whose primary objective is twofold: first, to encode the graph structure derived from the phase space, and second, to process this encoded representation to produce a sequence of predicted observations over time.

This work makes four key contributions: (i) we introduce **ChaoticFuzz**, a novel framework that transforms univariate time series into graph-structured data via phase space reconstruction and fuzzy clustering, enabling GNNs to effectively model chaotic and nonlinear temporal dynamics; (ii) we propose a new method for constructing a weighted adjacency matrix based on the similarity of fuzzy membership degrees, allowing the graph **to encode both proximity and uncertainty in the reconstructed phase space**; (iii) we integrate a global attention mechanism into the GNN architecture to jointly **capture spatial and temporal dependencies**, enhancing the model's ability to learn long-range patterns; and (iv) we demonstrate, through extensive experiments, that our approach consistently outperforms state-of-the-art forecasting models on chaotic time series, highlighting the benefits of **combining spatial trajectories with temporal evolution in a unified graph-based representation**.

## 2   RELATED WORK

Historically, univariate time series modeling has relied on statistical methods that aim to predict future behavior by identifying underlying patterns such as trend, seasonality, and noise. In this context, trend and seasonality are typically treated as deterministic components, while noise is considered stochastic. In statistics, one of the most widely used methods for time series modeling is the Seasonal Autoregressive Integrated Moving Average (SARIMA) model (Box et al., 2015; Shumway & Stoffer, 2011). SARIMA captures temporal dynamics by combining several components: an Autoregressive (AR) process that models dependencies on past observations, an integration (I) to remove trend, a Moving Average (MA) process that accounts for dependencies on past noise values, and seasonal (S) differencing to handle periodic patterns. The importance of modeling time series by explicitly capturing their trend and seasonal components was also thoroughly investigated by the DLinear model (Zeng et al., 2023), which has shown superior performance compared with recent, more complex networks.

In contrast to classical statistical methods specifically designed to capture time-series components, Artificial Neural Networks (ANNs) are architectures capable of learning patterns directly from the data. The adoption of traditional ANN-based approaches for modeling temporal data faces a fundamental challenge: they are specifically designed to handle i.i.d. data. A more suitable solution for modeling temporal dependencies is the use of Recurrent Neural Networks (RNNs), which are Deep Neural Networks (DNNs) that maintain a hidden state that allows information to persist over time, effectively learning and capturing sequential patterns. The most well-known RNN architectures are Long Short-Term Memory (LSTM) (Hochreiter & Schmidhuber, 1997) and Gated Recurrent Unit (GRU) (Cho et al., 2014), which incorporate a set of gating mechanisms to mitigate issues related to vanishing and exploding gradients during Backpropagation Through Time (BPTT) (Werbos, 1990). Regarding the adoption of attention mechanisms in DNNs, PatchTST (Nie et al., 2023) is a Transformer-based approach for time series forecasting and self-supervised representation learning by segmenting time series into subseries-level patches used as input tokens.

Recently, foundation models (FMs) have been applied to time-series modeling. In summary, FMs are large deep networks pre-trained on massive datasets to capture broad patterns and domain-general knowledge (Liang et al., 2024). A key advantage is that FMs can be efficiently adapted to new tasks using relatively small amounts of task-specific data. In this work, we analyze two FMs: CHRONOS (Ansari et al., 2024) and TimesFM (Das et al., 2024). In 2025, Zhang & Gilpin (2025) presented a study analyzing the success of FMs in capturing chaotic dynamics in time series.

From a chaos-theory perspective, recent DNN research has introduced mechanisms that leverage dynamical-systems structure to learn time-series features from reconstructed phase space, as in Attraos (Hu et al., 2024). While closely related to our approach, Attraos does not explicitly model spatial transitions between complex phase-space structures (e.g., attractors and fixed points), which our graph-based method does. By enabling the transformation of a univariate time series into a graph structure, as detailed in the following sections, we introduce a framework capable of capturing spatiotemporal patterns. This perspective not only enriches the representation of temporal dynamics but also leads to substantial improvements in forecasting performance, particularly in complex and chaotic settings.

## 3 PROBLEM DEFINITION

The process of unfolding time series into phase space was initially studied by Whitney (1936), who applied concepts from differential topology to embed functions into higher-dimensional spaces. In this context, Whitney proposed the immersion theorem, which states that a smooth manifold can be embedded in a higher-dimensional Euclidean space without self-intersections. This result laid the theoretical foundation for reconstructing attractors in phase space, enabling a clearer understanding of the underlying dynamics of chaotic systems. The definition of phase space is closely tied to the concept of fixed points, which support the reconstruction of attractors and orbits by revealing how time series observations evolve over time. To better understand this process, let $X_t = \{x_0, x_1, \ldots, x_t\}$ be a time series composed of $t$ observations, such that $x_i \in \mathbb{R}^d$ is a $d$-dimensional observation collected at time step $i$. Let $f$ be a map in $\mathbb{R}$ and $p$ a point such that $f(p) = p$. If all points in a neighborhood $\mathcal{N}$ around $p$ converge to $p$ under successive applications of $f$, then $p$ is called an attractive fixed point. Conversely, if all nearby points diverge from $p$, then $p$ is referred to as a repelling fixed point. Similarly, an orbit refers to a trajectory in phase space where observations are either attracted to or repelled from specific regions, such as fixed points or attractors. Building on Whitney's foundational work, Takens (1980) introduced his celebrated embedding theorem, which showed that it is possible to reconstruct the underlying dynamics of a system from time series observations. According to this theorem, a univariate time series $X_t$ can be embedded into a reconstructed phase space using time-delay coordinates, denoted by $\tilde{X}_{m,\tau}$. Each embedded observation is defined as $x_i(m, \tau) = \{x_i, x_{i+\tau}, \ldots, x_{i+(m-1)\tau}\}$, where $m$ is the embedding dimension and $\tau$ is the time delay (also referred to as the lag or separation parameter).

The embedding dimension fundamentally represents the minimum number of axes required to unfold a time series into phase space, allowing its underlying dynamics to be reconstructed without overlap. In contrast, the delay dimension (or time lag) plays a crucial role in capturing periodic structures or seasonality, as it determines the temporal separation between successive observations in the reconstructed space. The estimation of the appropriate embedding dimension was studied

by Takens (1980) and Mañé (1980), who established that the upper bound for the embedding dimension $D_e \in \mathbb{N}$ can be defined in terms of the fractal dimension $D_f$, such that $D_e > 2.0 \cdot D_f$. However, later work by Kennel et al. (1992) demonstrated that this theoretical bound often overestimates the necessary dimension, thereby increasing computational complexity and analysis time in practical phase space reconstructions.

To address this limitation, Kennel et al. (1992) proposed the False Nearest Neighbors (FNN) method, which evaluates the local neighborhood of each observation in phase space to determine the appropriate embedding dimension. The method begins with an initial embedding dimension of one and incrementally increases it, recalculating distances between neighboring points at each step. If the distance between a point and its neighbor grows significantly when projected into a higher-dimensional space, the neighbor is considered 'false', indicating that points appearing close in lower dimensions are actually distant in the true state space. This process continues until the proportion of false neighbors drops to zero (or below a predefined threshold), at which point the current embedding dimension is deemed sufficient for reconstructing the system's dynamics.

Formally, the FNN method considers an embedding dimension $m$, where the $r$-th nearest neighbor of an observation $x_i$ in phase space is denoted by $x_i^{(r)}$. The Euclidean distance between $x_i$ and its $r$-th neighbor in $m$ dimensions is computed as shown in Equation 1. When the embedding dimension is increased to $m + 1$, each observation vector $x_i$ is augmented with an additional coordinate, $x_{i+m\tau}$, resulting in the updated distance shown in Equation 2. This allows the method to quantify how neighborhood relationships evolve with increasing dimensionality, using the relative distance variation defined in Equation 3.

$$R_m^2(i, r) = \sum_{k=0}^{m-1} \left( x_{i+k\tau} - x_{i+k\tau}^{(r)} \right)^2 \tag{1}$$

$$R_{m+1}^2(i, r) = R_m^2(i, r) + \left( x_{i+m\tau} - x_{i+m\tau}^{(r)} \right)^2 \tag{2}$$

$$V_{i,r} = \sqrt{\frac{R_{m+1}^2(i, r) - R_m^2(i, r)}{R_m^2(i, r)}} = \frac{\left| x_{i+m\tau} - x^{(r)}i + m\tau \right|}{\sqrt{R_m^2(i, r)}} \tag{3}$$

According to Kennel et al. (1992), if the distance variation $V_{i,r}$ exceeds a predefined threshold $R_{\text{tol}}$, the corresponding neighbor is considered false. As discussed by the authors, a commonly accepted threshold is $R_{\text{tol}} \geq 10.0$. There are several methods available to estimate the delay parameter $\tau$. In this work, we adopt the approach proposed by Fraser & Swinney (1986), who introduced the Average Mutual Information (AMI) method for delay estimation. AMI evaluates the mutual dependence between time series values separated by varying time delays. A curve is constructed by plotting the average mutual information against different delay values, and the first local minimum of this curve is selected as the optimal delay $\tau$.

The first two plots in Figure 1 illustrate the transformation of a time series into its corresponding phase space. The first plot ("Time Series") displays observations generated by the Lorenz system (described later), while the second ("Phase Space") shows the reconstructed phase space using an embedding dimension of $m = 3$ and a time delay of $\tau = 2$.

The reconstruction of time series in phase space with FNN and AMI opens up new possibilities for modeling complex dynamics. By revealing the underlying structure of trajectories and highlighting patterns through clusters of similar observations, this approach allows us to move beyond traditional time-based representations. Inspired by this potential, we propose a novel framework that transforms the unfolded time series into a graph-based representation, enabling rich temporal and spatial modeling of chaotic systems.

## 4 METHODOLOGY

The graph-based construction aims to capture both the structural and temporal dynamics revealed by the phase space representation. To achieve this, we employ a soft clustering strategy based on

Fuzzy systems (Zadeh, 1965), which allows observations to belong to multiple clusters with varying degrees of membership.

**Fuzzy Clustering.** To address limitations of traditional (crisp) clustering algorithms, in which each data point is assigned to exactly one cluster, Fuzzy clustering relaxes this constraint by allowing each data point to belong to multiple clusters with varying degrees of membership (Bezdek, 1981). This is particularly useful when cluster boundaries are ambiguous or overlapping. Furthermore, membership values can reveal more nuanced relationships between data points and clusters, enabling a deeper understanding of the underlying structure (Xu & Wunsch, 2008). In our framework, we employ Fuzzy C-Means (FCM) (Bezdek, 1981), which was developed to minimize the cost function presented in Equation 4, where $J$ denotes the objective function to be minimized, $x_i$ is the $i$-th data point, $c_j$ is the center of the $j$-th cluster, $u_{ij} \in [0, 1]$ represents the degree of membership of $x_i$ in cluster $c_j$, $f > 1$ controls the level of cluster fuzziness, $N$ is the number of data points, and $C$ is the number of clusters. This minimization process must also satisfy the following constraints: $\sum_{j=1}^{C} u_{ij} = 1, \forall j$, and $0 < \sum_{i=1}^{N} u_{ij} < N, \forall i$.

$$J = \sum_{j=1}^{C} \sum_{i=1}^{N} u_{ij}^f \cdot ||x_i - c_j||^2 \qquad (4)$$

The optimization process in Fuzzy C-Means is performed iteratively by updating the membership degrees $u_{ij}$ and the cluster centroids $c_j$ until convergence. First, the membership degree $u_{ij}$ of a data point $x_i$ in cluster $j$ is computed based on its distance to all cluster centroids:

$$u_{ij} = \frac{1}{\sum_{k=1}^{C} \left( \frac{|x_i - c_j|}{|x_i - c_k|} \right)^{\frac{2}{f-1}}} \qquad (5)$$

Then, the centroid $c_j$ of cluster $j$ is updated using the weighted average of all data, where the weights are the membership degrees raised to the fuzziness parameter $f$:

$$c_j = \frac{\sum_{i=1}^{N} u_{ij}^f \cdot x_i}{\sum_{i=1}^{N} u_{ij}^f} \qquad (6)$$

This iterative process continues until a stopping criterion is met, such as when the change in the objective function $J$ falls below a predefined threshold or a maximum number of iterations is reached.

**Graph Structure**. The main contribution of this work builds upon three well-established areas with strong formal and theoretical foundations: Chaos Theory, Fuzzy Systems, and Graph Neural Networks. Tools from Chaos Theory are employed to unfold a time series into its phase space, making it possible to reveal spatial and temporal structures such as attractors and trajectories. Complementarily, fuzzy clustering is used to identify groups of observations with similar behavior by assigning them to clusters with varying degrees of membership. These spatial relationships and behavioral similarities are then leveraged to construct graph structures that serve as the input for GNN-based predictive modeling.

In this sense, we define $G = (V, E, \tilde{X}_{m,\tau})$ as an attributed, undirected graph, where $V$ and $E$ represent the sets of nodes (vertices) and edges, respectively, and $\tilde{X}_{m,\tau}$ denotes the set of phase-space observations, which serve as attribute vectors associated with the nodes in $V$. Fuzzy logic is employed in two distinct stages during the construction of $G$. Initially, Fuzzy is used to compute a membership matrix $U = [u_{ij}] \in [0, 1]^{N \times C}$, where $u_{ij}$ represents the degree of membership of observation $x_i(m, \tau)$ to cluster $C_j$. In the first stage, each observation is strictly assigned to the cluster with the highest membership degree, i.e., $x_i(m, \tau) \in C_j$ such as $j = \arg\max_{k \in \{1, ..., C\}} u_{ik}$, as illustrated in the bottom plot of the "Fuzzy Clustering" block in Figure 1. Each resulting cluster is then considered as a node in the graph, forming the set $V$.

In the second stage, the fuzzy membership matrix $U$ is used to construct weighted edges between the nodes of the graph. Specifically, the phase-space observations belonging to each pair of clusters are used to quantify the connection strength between their corresponding nodes. This strategy enables

the edge weights to reflect the degree of interaction or similarity between clusters based on shared or overlapping membership patterns. To better illustrate this process, consider the top plot of the "Fuzzy Clustering" block in Figure 1, which shows the varying membership values (represented by different colors) assigned to each cluster for all phase-space observations.

The edge weight between clusters $j$ and $l$, where $j, l \in \{1, ..., C\}$, is defined by Equation 7. For each phase-space observation, denoted as $x_i$ for simplicity, the term inside the maximization computes a similarity score based on the arithmetic mean of the membership degrees $u_{ij}$ and $u_{il}$, penalized by their absolute difference. This formulation favors connections between clusters for which observations simultaneously exhibit high and similar degrees of membership, thereby reflecting stronger inter-cluster relationships.

$$A_{jl} = \max_{x_i \in \bar{X}_{m,\tau}} \left( \frac{u_{ij} + u_{il}}{2} - |u_{ij} - u_{il}| \right) \tag{7}$$

The "Graph Neural Network" block in Figure 1 illustrates the graph structure according to our proposed methodology. The color-shaded dots represent the vertex attributes, corresponding to the phase-space observations, while the black edges indicate strong connections between clusters. In contrast, the lighter (shaded) edges represent weaker relationships based on the computed edge weights.

Univariate time series have also been modeled using Horizontal Visibility Graphs (HVGs) (Luque et al., 2009), which map observations to nodes and connect them through local visibility rules. While HVGs provide a useful topological representation of time series, their focus on local visibility relations limits their ability to capture the long-range dependencies needed for accurate phase space reconstruction in chaotic systems.

**Model Architecture.** After transforming univariate time series into spatiotemporal representations, we can effectively model both the observations and their underlying relationships using GNNs. This graph-based approach allows the architecture to capture complex temporal dynamics and structural dependencies that are often overlooked by traditional sequential models. The execution of GNNs depends on encoder-decoder functions to represent the graph as node embeddings, which is processed by using Neural Message Passing (NMP). In each message-passing iteration performed during the training, new knowledge from node embeddings is updated according to information aggregated from their neighborhoods (Hamilton, 2020; Wu et al., 2021).

During each message-passing iteration in a GNN, a hidden embedding $\mathbf{h}_v^k$ corresponding to each node $v \in V$ is updated according to information aggregated from $v$'s graph neighborhood $\mathcal{N}(v)$ (Hamilton, 2020). This message-passing update can be expressed by Equations (8) and (9).

$$\mathbf{h}_v^k = UPDATE^k \left( \mathbf{h}_v^{k-1}, \mathbf{m}_{\mathcal{N}(v)}^k \right) \tag{8}$$

$$\mathbf{m}_{\mathcal{N}(v)}^k = AGGREGATE^k \left( \left\{ \mathbf{h}_u^k, \forall u \in \mathcal{N}(v) \right\} \right) \tag{9}$$

From these equations, consider UPDATE and AGGREGATE as Neural Networks. At each iteration $k$ of the GNN, the AGGREGATE function takes as input the set of embeddings of the nodes in $v$'s graph neighborhood $\mathcal{N}(v)$ and generates a message $\mathbf{m}_{\mathcal{N}(v)}$ based on this aggregated neighborhood information. The different iterations of message passing are also sometimes known as the different layers of the GNN (Hamilton, 2020). The function UPDATE then combines the message $\mathbf{m}_{\mathcal{N}(v)}^k$ with the previous embedding $\mathbf{h}_v^{k-1}$ of node $v$ to generate the updated embedding $\mathbf{h}_v^k$. The initial embeddings at $k = 0$ are set to the input features for all the nodes, i.e., $\mathbf{h}_v^{(0)} = \mathbf{x}_v, \forall v \in V$. After running $K$ iterations of the GNN message passing, we can use the output of the final layer to define the embeddings for each node.

At this stage, we evaluated several GNN architectures, as listed in the "Baseline Models" section. To maintain a concise and focused discussion on our proposed contribution, detailed descriptions of these baseline models are provided in the appendices. Beyond enabling the spatiotemporal modeling of univariate time series, a key contribution of this work is the design of a novel GNN-based architecture, depicted in Figure 2. Our architecture begins with a graph neural network model, to extract

informative representations from the phase-space graph. Specifically, GNN layers are employed to generate node embeddings based on our custom adjacency matrix (Equation 7), which encodes fuzzy-based similarities between phase-space regions.

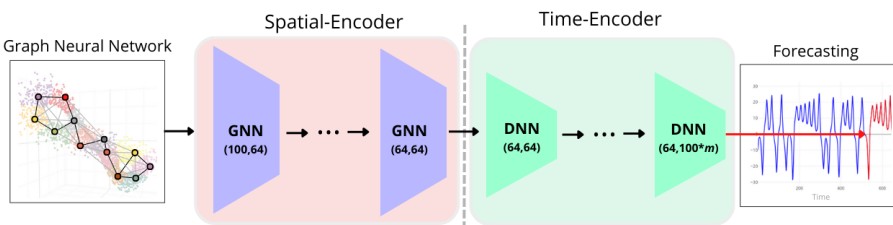

Figure 2: Phase-Space Encoder-Encoder architecture.

Using an *encoder-encoder* strategy, the data for generating node embeddings were extracted while preserving the trajectory in the phase space. This approach is especially important for capturing the chaotic dynamics inherent in time series data. Spatial information is *encoded* with different GNN layers, while temporal information is *encoded* with DNNs, specifically MLPs. In our experiments, GraphSAGE yielded the best overall results. We then replaced the MLP *time encoder* with a sequence comprising an LSTM, global attention, and an MLP.

## 5 EXPERIMENTS

**Datasets.** To assess ChaoticFuzz, we conducted experiments using five well-known chaotic time series (Alligood et al., 1997; Swiercz, 2006): (i) Lorenz, (ii) Rössler, (iii) Logistic, (iv) Hénon, and (v) Chua. The Lorenz time series is generated by the following system of ordinary differential equations shown: $\frac{dx}{dt} = \sigma(y-x)$; $\frac{dy}{dt} = x(\rho-z)-y$; and $\frac{dz}{dt} = xy-\beta z$. The system was originally proposed to model atmospheric convection phenomena. For our experiments, the parameters were set to $\sigma = 10$, $\beta = \frac{8}{3}$, and $\rho = 28$, ensuring the system exhibits chaotic behavior.

The Rössler time series is derived from the following system of differential equations: $\frac{dx}{dt} = -x-y$; $\frac{dy}{dt} = x + ay$; and $\frac{dz}{dt} = b + z(x-c)$. It was originally proposed to model chemical turbulence. In our experiments, the parameters were set to $a = 0.2$, $b = 0.2$, and $c = 5.7$, which are known to produce chaotic behavior in the system. The logistic time series is governed by $x_{n+1} = rx_n(1-x_n)$ and exhibits chaotic behavior when initialized with $x_n = 0.5$ and $r = 3.8$. The last time series we utilized was derived from the Hénon map, which is based on the equations: $x_{n+1} = 1 - ax_n^2 + y_n$; and $y_{n+1} = bx_n$. Although this map is known for its chaotic behavior across different parameter sets, our specific series were generated using $a = 1.4$, $b = 0.3$. The Chua time series is generated by the following system of ordinary differential equations: $\frac{dx}{dt} = \alpha(y - x - h(x))$; $\frac{dy}{dt} = x - y + z$; $\frac{dz}{dt} = -\beta y$, where $h(x)$ is a piecewise-linear function that introduces nonlinearity into the system. This system was originally proposed to describe an electronic circuit capable of producing chaotic signals. For our experiments, the parameters were set to $\alpha = 15.6$, $\beta = 28$, $m_0 = -1.143$, and $m_1 = -0.714$, ensuring the system exhibits chaotic behavior.

**Evaluation Protocol.** The evaluation followed a standard procedure commonly used in time series analysis. Each time series consisted of 2,000 observations, with the first 1,600 used for training all models. From the remaining 400 observations, a sliding window approach was applied: the first 100 were used as input to predict the next 100, and this process was repeated to produce a total of 300 predictions.

In time series analysis, several regression metrics have been proposed in the literature. In this work, for the sake of conciseness, we adopt the Root Mean Squared Error (RMSE), as it is widely used in similar studies and is also commonly employed as a loss function during the training phase. However, as discussed in our experiments, traditional error metrics compute pairwise differences between predicted and true values. As a result, they tend to penalize models that attempt to follow the underlying time series dynamics more heavily than those that simply predict a constant mean value, despite the latter being less informative in capturing the system's actual behavior.

This problem is addressed using the Dynamic Time Warping (DTW) distance, which seeks the optimal alignment between two time series before computing their dissimilarity (Ding et al., 2008). DTW is particularly effective for comparing time series that may be out of phase or exhibit temporal distortions, as it allows non-linear warping along the time axis.

**Baseline Models.** The evaluation of our proposed method was carried out in comparison with several baseline models, previously discussed in the Related Work section, that are widely used for time series forecasting: SARIMA, DLinear, LSTM, GRU, PatchTST, CHRONOS, TimesFM, and Attraos. For TimesFM, we considered two model variants distinguished by their parameter size: 200 million and 500 million parameters. After constructing the graph representations, we evaluated our proposed framework using several Graph Neural Network (GNN) architectures, including GCN (Kipf & Welling, 2017), SAGE (Hamilton et al., 2017), CHEB (Defferrard et al., 2016), GAT (Veličković et al., 2017), and LEConv (Ranjan et al., 2020).

**Hyperparameters.** We report complete hyperparameter configurations in the appendices, due to the large set of state-of-the-art models and page limits. Data and code repository: `https://anonymous.4open.science/r/ChaoticFuzz-5505/README.md`.

## 6 RESULTS

We evaluate our approach on long-term forecasting for several chaotic time series and compare it against strong baselines (Table 1). In the DTW columns, boldface marks the best score in each column; blue cells indicate the best score achieved by our method, and red cells indicate the best score among SOTA baselines. Figure 3 provides visual inspection: the leftmost panel plots ground truth (black curve) and predictions (blue curve = ours; red curve = best SOTA from Table 1) used to compute RMSE. The next two panels show DTW alignment paths for our method (middle) and the SOTA baseline (right). Alignment paths closer to the diagonal, and with smaller enclosed area, indicate better temporal correspondence.

| Approach | Model | Lorenz | | Hénon | | Rössler | | Logistic | | Chua | |
|---|---|---|---|---|---|---|---|---|---|---|---|
| | | DTW | RMSE | DTW | RMSE | DTW | RMSE | DTW | RMSE | DTW | RMSE |
| | SARIMA | 3.382 | 12.847 | 0.433 | 1.019 | 0.707 | 2.978 | 0.105 | 0.247 | 0.444 | 1.401 |
| | DLinear | 4.367 | 12.397 | 0.455 | 1.028 | 0.463 | 2.422 | 0.109 | 0.250 | 0.419 | 1.379 |
| | LSTM | 2.801 | 13.101 | 0.442 | 1.027 | 0.352 | 1.847 | 0.107 | 0.249 | 0.313 | 1.672 |
| | GRU | 3.863 | 14.087 | 0.412 | 1.099 | 0.132 | 1.110 | 0.108 | 0.250 | 0.280 | 1.823 |
| SOTA | PatchTST | 5.604 | 20.686 | 0.445 | 1.073 | 1.872 | 7.030 | 0.109 | 0.250 | 0.365 | 1.519 |
| | CHRONOS | 3.650 | 12.016 | 0.445 | 1.073 | 1.670 | 9.869 | 0.108 | 0.259 | 0.420 | 1.444 |
| | TimesFM_200 | 3.854 | 11.994 | 0.440 | 1.038 | 2.282 | 12.619 | 0.106 | 0.249 | 0.472 | 1.314 |
| | TimesFM_500 | 3.796 | 12.349 | 0.439 | 1.065 | 1.588 | 10.088 | 0.107 | 0.256 | 0.298 | 1.554 |
| | Attraos | 4.804 | 15.517 | 0.454 | 1.018 | 1.957 | 7.125 | 0.111 | 0.274 | 0.361 | 1.451 |
| | GCN | 3.958 | 13.766 | **0.412** | 1.166 | 1.375 | 4.523 | **0.104** | 0.287 | **0.065** | 0.212 |
| | SAGE | 5.333 | 13.764 | **0.404** | 1.275 | 1.646 | 5.536 | **0.101** | 0.347 | **0.058** | 0.207 |
| ChaoticFuzz | CHEB | **2.729** | 13.686 | 0.428 | 1.149 | 1.581 | 10.029 | **0.103** | 0.303 | **0.061** | 0.214 |
| | GAT | 4.177 | 14.696 | 0.419 | 1.165 | 3.272 | 13.428 | **0.105** | 0.295 | **0.061** | 0.209 |
| | LECONV | 2.838 | 13.469 | 0.432 | 1.273 | 1.079 | 5.096 | **0.104** | 0.312 | **0.068** | 0.194 |
| | Global+SAGE | **2.513** | 15.258 | 0.425 | 1.306 | 0.551 | 4.657 | 0.128 | 0.362 | **0.059** | 0.188 |

Table 1: Evaluation results highlighting the importance of using spatiotemporal information extracted from univariate time series. DTW scores emphasize each model's ability to follow the underlying dynamics.

We begin by analyzing the Lorenz system, a standard benchmark for chaotic time-series modeling due to its sensitivity to initial conditions and complex attractor structure. Our framework, Chaotic-Fuzz, attains the lowest DTW score across deep neural networks, foundation models, and classical statistical baselines, highlighting the value of combining phase-space reconstruction with graph-based representations and global attention. Although our RMSE is higher than some DNN baselines, the DTW results indicate that ChaoticFuzz more faithfully tracks the system's trajectory rather than merely minimizing pointwise error. Figure 3(a) illustrates this: ChaoticFuzz follows sharp spikes and local oscillations, whereas the best baseline (LSTM) tends to regress toward a smoothed average, missing key chaotic dynamics.

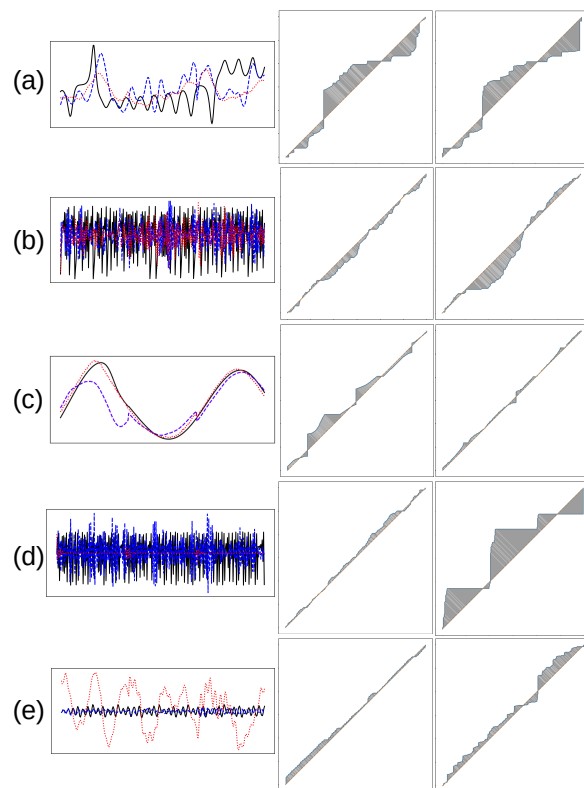

Figure 3: Comparing different models in time domain.

Similar trends hold for the Hénon, Logistic, and Chua series. ChaoticFuzz attains the best results across all three, underscoring the adaptability of GNN variants to distinct chaotic regimes. As illustrated in Figures 3(b), (d), and (e), our method faithfully captures the oscillatory patterns of the ground truth, whereas the baselines – GRU in (b) and (e), and SARIMA in (d) – fail to reproduce these fluctuations. Notably, for the Logistic and Chua series, the top-performing baseline collapses toward the mean, yielding deceptively low RMSE but poor dynamical fidelity (as reflected by DTW), and thus missing the true system behavior.

A noteworthy exception arises with the Rössler series. In this case, our model does not outperform the GRU baseline. As shown in Figure 3(c), the ground truth (black) exhibits a relatively simple, near-sinusoidal pattern that the GRU (red) tracks with high precision. By contrast, ChaoticFuzz, designed to uncover richer phase-space structure, appears to model dynamics that are not present, slightly degrading performance. This suggests that when a series lacks strong chaotic behavior or complex attractor structure, the advantages of our approach may diminish. We include this result to provide a balanced assessment, highlighting both the strengths and the limitations of our method across different dynamical regimes.

## 7  CONCLUSION

We presented a novel architecture that fuses chaos-theory principles, fuzzy clustering, and graph neural networks to model univariate time series via phase-space reconstruction. By converting latent spatiotemporal structure into a graph-based representation, the method captures both local temporal dependencies and global dynamical structure, yielding more accurate and robust long-term forecasts. Empirically, we observe consistent gains, including superior DTW scores, indicating stronger trajectory preservation in chaotic regimes, not just pointwise performance. These results suggest a generalizable framework for forecasting nonlinear, high-dimensional, and chaotic processes from simple univariate observations, and point to promising avenues for extending spatial reasoning and phase-space modeling in future work.

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

# A   BASELINES

## A.1   DECOMPOSITION MODELS

Decomposition-based approaches explicitly separate time series into components such as trend, seasonality, and residual fluctuations before modeling. This strategy allows models to focus on distinct temporal patterns, improving interpretability and predictive accuracy. In this subsection, we present both classical statistical techniques, such as SARIMA (Box et al., 2015), and modern neural approaches, such as DLinear (Zeng et al., 2023), which extend the decomposition principle to enhance long-term forecasting performance.

**SARIMA.** Historically, univariate time series modeling has relied on statistical methods that aim to predict future behavior by identifying underlying patterns such as trend, seasonality, and noise. In this context, trend and seasonality are typically treated as deterministic components, while noise is considered stochastic. In statistics, one of the most widely used methods for time series modeling is the Seasonal Autoregressive Integrated Moving Average (SARIMA) model Box et al. (2015); Shumway & Stoffer (2011). SARIMA captures temporal dynamics by combining several components: an Autoregressive (AR) process that models dependencies on past observations, an integration (I) to remove trend, a Moving Average (MA) process that accounts for dependencies on past forecast errors, and seasonal (S) differencing to handle periodic patterns. This flexible framework allows to model a wide range of time series behaviors, including trends, cycles, and seasonal effects.

Time series can be modeled in terms of $q$ past random observations as Moving Average process, MA($q$), $x_t = x_{t-1} + \theta_1 \cdot \varepsilon_{t-1} + \cdots + \theta_q \cdot \varepsilon_{t-q}$, such that $\{\theta_q\}$ are constants and $\{\varepsilon_t\}$ are values produced by a purely random process. Time series can also be modeled by $p$ past observations as an Autoregressive process, AR($p$), $x_t = \phi_1 \cdot x_{t-1} + \cdots + \phi_p \cdot x_{t-p} + \varepsilon_t$, such that $\{\phi_p\}$ are constants and $\varepsilon_t$ is value produced by a purely random process.

These processes can also be combined to model time series according to past noise values and observations using an Autoregressive and Moving Average (ARMA) process. The seasonal ARIMA model, referred to as SARIMA, was developed to use differencing at a lag equal to the number of seasons Shumway & Stoffer (2011), aiming to remove seasonal effects, as shown in $\Phi_P(B^s)\phi(B) \bigtriangledown_s^D \bigtriangledown^d x_t = \delta + \Theta_Q(B^s)\theta(B)\varepsilon_t$.

**DLinear** (Zeng et al., 2023) is a neural forecasting model that combines a decomposition scheme with linear transformations. The method first separates the raw time series into two components: a trend, extracted using a moving average kernel, and a remainder (seasonal) component. Each component is then passed through an independent one-layer linear transformation, and the results are summed to form the final prediction. By explicitly isolating and modeling the trend, DLinear enhances the predictive power of simple linear baselines, especially in settings where long-term directional patterns dominate. This design challenges the assumption that increasingly complex Transformer-based architectures are necessary for long-term forecasting, showing instead that decomposition combined with linear modeling can achieve competitive, and often superior, results.

## A.2   DEEP NEURAL NETWORK MODELS

Deep Neural Network (DNN) models have emerged as powerful alternatives to classical statistical approaches for time series forecasting. In this subsection, we highlight representative architectures ranging from recurrent models, such as RNNs and their variants, to more recent Transformer-based approaches like PatchTST (Nie et al., 2023), which introduce new design strategies tailored to temporal data.

**RNN.** In contrast to classical statistical methods specifically designed to capture time-series components, Artificial Neural Networks (ANNs) are architectures capable of learning patterns directly from the data. The adoption of traditional ANN-based approaches for modeling temporal data faces a fundamental challenge: they are specifically designed to handle i.i.d. data. A more suitable solution for modeling temporal dependencies is the use of Recurrent Neural Networks (RNNs), which are Deep Neural Networks (DNNs) that maintain a hidden state that allows information to persist over time, effectively learning and capturing sequential patterns. The most well-known RNN architecture is the LSTM network Hochreiter & Schmidhuber (1997) and the Gated Recurrent Unit

(GRU) Cho et al. (2014), which incorporates a set of gating mechanisms to mitigate issues related to vanishing and exploding gradients during Backpropagation Through Time (BPTT) Werbos (1990).

**PatchTST.** Recent advances in Transformer architectures have also been adapted to the time series forecasting domain. A representative example is the PatchTST model (Nie et al., 2023), which introduces a channel-independent patch-based design. In this approach, each variable of a multivariate series is treated as an independent univariate sequence. Although the channels share the same Transformer backbone, their forward computations are performed independently, which allows the model to capture detailed temporal dynamics for each variable without interference from others.

Prior to entering the Transformer encoder, each univariate sequence is normalized through an instance normalization operator and then segmented into patches. These patches serve as input tokens, reducing sequence length while preserving local temporal structure. Additionally, PatchTST incorporates masked self-supervised learning, in which randomly selected patches are masked and reconstructed by the model. This strategy enhances representation quality and improves the robustness of the learned embeddings, making the model particularly effective for long-term forecasting tasks.

### A.3 FOUNDATION MODELS

These models are trained on massive and diverse datasets, enabling them to capture general temporal representations that can be adapted to a wide range of forecasting tasks with little or no fine-tuning. Unlike traditional architectures designed specifically for individual datasets, Foundation Models (FMs) emphasize scalability, transferability, and zero-shot capabilities. In this subsection, we present two representative approaches: CHRONOS (Ansari et al., 2024), which adapts language modeling techniques to time series by tokenizing observations, and TimesFM (Das et al., 2024), a Transformer-based foundation model directly pretrained on large-scale temporal data.

**CHRONOS.** An important FM is CHRONOS (Ansari et al., 2024), which is a framework that considers advances in language model architectures and training methodologies to address the challenges of time series forecasting. Although both domains share a sequential nature, they diverge in terms of data representation. Natural language comprises discrete tokens from a predefined vocabulary, while time series are characterized by continuous-valued observations. Therefore, CHRONOS aims to bridge both areas by transforming time series into sequences of tokens, as words, that can later be analyzed by LLMs. The forecasts produced by CHRONOS are based on probabilistic models, which allow them to generate multiple possible future scenarios by sampling repeatedly from their predicted range. The predicted token is dequantizated, thus mapping back to real values and unscaled to obtain the real forecast.

Language models operate on tokens from a finite vocabulary, so using them for time series data requires mapping the observations to a finite set of tokens. To this end, CHRONOS first scale and then quantize observations into a fixed number of bins. This function selects $B$ bin centers and defines intervals considering $c_1 < \cdots < c_B$, i.e., $q(X_t) : \mathbb{R} \rightarrow \{1, 2, \ldots, B\}$. After transforming the time-series observations into tokens, CHRONOS uses a vocabulary, $|\mathcal{V}_{ts}|$, to pad the time series including a sequence of PAD characters, keeping all series to the same size, and add an EOS character to mark the actual last observation. After this transformation, the resultant time series is processed by LLM as a normal sequence of characters $\mathbf{Z}_t = \{z_1, \ldots, z_t\}$.

The training process relies on the loss function presented in Equation 10, which minimizes the outputs of a model parameterized by $\theta$. This model learns from a context window $C$ of a tokenized time series $z_t$ and predicts future observations over a horizon $H$. As evident from the structure of the loss function, CHRONOS performs regression through classification.

$$
\ell(\theta) = - \sum_{h=1}^{H+1} \sum_{i=1}^{|\mathcal{V}_{ts}|} \mathbf{1}_{(z_{C+h+1}=i)} \log p_\theta(z_{C+h+1} = i \mid \mathbf{z}_{1:C+h}) \tag{10}
$$

**TimesFM.** From a different perspective, instead of modeling time series as sequences of words, TimesFM (Time-Series Foundation Model) (Das et al., 2024) is a large pretrained model directly trained on massive amounts of time-series data to automatically learn temporal patterns. TimesFM is a transformer-based foundation model that has shown great zero-shot forecasting potential, achiev-

ing accuracy comparable to fully-supervised forecasting models. During the training phase, one of the core concepts employed in the TimesFM architecture is the patching of time series data to enhance inference speed and support long-term forecasting. For the patches, TimesFM generates corresponding masks to hide specific observations during training. This mechanism is designed to accommodate varying context lengths, ensuring robust predictive performance even when the input length is not an exact multiple of the patch size. Each patch–mask pair is processed by a residual block composed of multilayer perceptrons with skip connections. The output is then combined with positional encodings to indicate the patch's location within the original time series. TimesFM implements stacked layers using standard multi-head self-attention mechanisms followed by feed-forward networks. TimesFM is seen as a decoder-only model that performs autoregressive predictions, generating the next patch based on the given context and previously forecasted patches.

### A.4 CHAOTIC MODELS

Chaotic models leverage principles from chaos theory to capture complex and nonlinear dependencies in time series data. By reconstructing the underlying dynamical system in phase space, these approaches exploit the structure of chaotic attractors to improve long-term forecasting performance.

In this work, we compare our proposed method with **Attraos** (Hu et al., 2024), a recent chaos-inspired model designed to preserve dynamical properties such as sensitivity to initial conditions and multiscale variability, which are essential for representing real-world temporal processes. To achieve this, it employs phase space reconstruction, which applies coordinate delay embedding to generate high-dimensional trajectories from observed sequences. This reconstruction does not require prior knowledge of the underlying system and is controlled by two hyperparameters: embedding dimension and time delay.

After reconstruction, Attraos applies a non-overlapping patching strategy to reduce model complexity and accelerate convergence. The resulting patches are organized into tensors, representing different dynamical windows of the system. This representation preserves the chaotic attractor structure, enabling the model to capture long-term dependencies. By adopting a channel-independent approach, the method unifies multiple variables into a coherent dynamical system, leveraging differences in Lyapunov exponents across series and improving robustness in multivariate forecasting.

### A.5 GRAPH NEURAL NETWORK MODELS

To analyze Graph Neural Networks (GNNs) in our research, we consider an input graph $G$ with node features $\mathbf{X} \in \mathbb{R}^{d \times |V|}$. From this representation, the model generates node embeddings $\mathbf{z}_v$ for all $v \in V$.

The functioning of GNNs relies on encoder–decoder architectures that learn node embeddings through *Neural Message Passing* (NMP). During training, each message-passing iteration updates node embeddings by aggregating information from their local neighborhoods (Hamilton, 2020; Wu et al., 2021).

Formally, at iteration $k$, the hidden embedding $\mathbf{h}_v^k$ of a node $v \in V$ is updated according to:

$$\mathbf{h}_v^k = \text{UPDATE}^{(k)}\Big(\mathbf{h}_v^{k-1}, \ \text{AGGREGATE}^{(k)}\{\mathbf{h}_u^{k-1} : u \in \mathcal{N}(v)\}\Big),$$

where $\text{AGGREGATE}^{(k)}(\cdot)$ collects information from the neighborhood of $v$, and $\text{UPDATE}^{(k)}(\cdot)$ combines this information with the node's previous representation.

In this work we have used Graph Convolutional Networks (GCNs), SAmple and aggreGatE (SAGE), Chebyshev spectral graph convolutional operator (CHEB), Graph Attention Networks (GAT), and Local Extrema Convolution (LEConv), described next.

**GCN.** Graph Convolutional Networks (GCNs) extend the message passing framework by introducing a convolution-like operator on graphs. A key property is weight-sharing, where the same parameter matrix is applied across all nodes, combined with symmetric normalization and the addition of self-loops. This enables scalable representation learning even on large graphs (Kipf & Welling, 2017).

The symmetric-normalized aggregation is defined as:

$$\mathbf{m}_{\mathcal{N}(v)} = \sum_{u \in \mathcal{N}(v)} \frac{\mathbf{h}_u}{\sqrt{|\mathcal{N}(v)| \, |\mathcal{N}(u)|}}, \tag{11}$$

where the contribution of each neighbor is normalized by the degrees of both nodes.

In practice, aggregation is performed over the neighborhood of $v$ including the node itself, effectively merging the update into the aggregation step. The GCN update rule is thus expressed as:

$$\mathbf{h}_v^k = \sigma \left( \mathbf{W}^k \sum_{u \in \mathcal{N}(v) \cup \{v\}} \frac{\mathbf{h}_u}{\sqrt{|\mathcal{N}(v)| \, |\mathcal{N}(u)|}} \right), \tag{12}$$

where $\mathbf{W}^k$ is the trainable weight matrix at layer $k$ and $\sigma$ is a nonlinear activation, typically ReLU.

**SAGE.** SAmple and aggreGatE (GraphSAGE) is an inductive GNN architecture that learns aggregation functions rather than storing explicit embeddings for each node (Hamilton et al., 2017). These functions combine local neighborhood information at different depths, enabling generalization to unseen nodes.

Let $AGGREGATE^{(k)}$ denote the aggregator at layer $k$ and $\mathbf{W}^k$ the trainable weight matrix. Given input features $\mathbf{x}_v$ for each node $v \in V$, the hidden representation at depth $k$ is updated in two steps. First, each node aggregates the representations of its neighbors:

$$\mathbf{h}_{\mathcal{N}(v)}^k = AGGREGATE^{(k)} \left( \left\{ \mathbf{h}_u^{k-1} : u \in \mathcal{N}(v) \right\} \right). \tag{13}$$

Typical aggregation operators include MEAN, SUM, or MAX.

Next, the node's previous representation is concatenated with the aggregated neighborhood vector and passed through a nonlinear transformation:

$$\mathbf{h}_v^k = \sigma \left( \mathbf{W}^k \cdot CONCAT \left( \mathbf{h}_v^{k-1}, \mathbf{h}_{\mathcal{N}(v)}^k \right) \right), \tag{14}$$

where $\sigma$ is an activation function such as ReLU.

**CHEB.** In another direction, the Chebyshev spectral graph convolutional operator (CHEB) (Defferrard et al., 2016) provides an efficient generalization of CNNs to arbitrary graphs by expressing convolutional filters as polynomials of the graph Laplacian $\mathbf{L}$.

Given an input signal $\mathbf{X} \in \mathbb{R}^N$ defined on a graph $G$ with $N$ nodes, the graph convolution of $\mathbf{X}$ with filter $g$ is:

$$\mathbf{X} *_{\mathcal{G}} g = p_M(\mathbf{L})\mathbf{X}, \tag{15}$$

where $p_M(\mathbf{L})$ is the spectral filter defined as a polynomial of degree $M$ in the Laplacian eigenvalues.

As noted by Hamilton (2020), using a polynomial of degree $M$ ensures that the filtered representation at each node incorporates information from its $M$-hop neighborhood. In CHEB, the filter $p_M(\mathbf{L})$ is parameterized using Chebyshev polynomials (Mason & Handscomb, 2002), which enables fast recursive computation and avoids the need for explicit eigenvalue decomposition.

**GAT.** Graph Attention Networks (GAT) (Veličković et al., 2017) extend the message passing framework by incorporating attention mechanisms to weight the importance of neighbors during aggregation. Instead of treating all neighbors equally, GAT computes coefficients that determine how much influence each neighbor has on the target node.

Formally, the representation of node $v$ after an attention-based aggregation is:

$$\mathbf{x}'_v = \sum_{u \in \mathcal{N}(v) \cup \{v\}} \alpha_{(v,u)} \mathbf{W} \mathbf{x}_u, \tag{16}$$

where $\mathbf{W}$ is a trainable linear transformation, and $\alpha_{(v,u)}$ denotes the normalized attention coefficient that quantifies the contribution of neighbor $u$ when updating node $v$.

To stabilize learning, multiple attention functions (*heads*) can be applied in parallel, each producing a different set of coefficients. Their outputs are then concatenated or averaged, in a manner similar to the multi-head mechanism in Transformer architectures (Vaswani et al., 2017).

**LECONV.** Local Extremum Convolution (LECONV) (Ranjan et al., 2020) extends the graph convolution framework by explicitly modeling local maxima and minima through a difference operator. This allows the network to highlight how a node's features deviate from those of its neighbors, thereby capturing extremal patterns in the graph.

The update rule is defined as:

$$\mathbf{x}'_v = \mathbf{x}_v \mathbf{W}_1 \; + \sum_{u \in \mathcal{N}(v)} e_{(u,v)} \cdot \left( \mathbf{W}_2 \mathbf{x}_v - \mathbf{W}_3 \mathbf{x}_u \right), \tag{17}$$

where $e_{(u,v)}$ is the edge weight between $u$ and $v$, and $\mathbf{W}_1, \mathbf{W}_2, \mathbf{W}_3$ are trainable parameter matrices.

By incorporating relative differences into the aggregation, LECONV emphasizes extremal structural and feature variations, making it well-suited for tasks where local contrast is informative.

## B  ADDITIONAL RESULTS

In this section, we present a set of complementary experiments and visual analyses to illustrate the superior performance of our proposed approach. As discussed in our manuscript, traditional error metrics compute pairwise differences between predicted and ground truth values. Consequently, they often penalize models that strive to capture the true dynamics of the time series more than those that simply approximate a static mean. To address this limitation, we employ Dynamic Time Warping (DTW), a distance metric that identifies the optimal temporal alignment between two sequences before calculating their dissimilarity.

A key advantage of DTW lies in its ability to provide not only a quantitative error measure but also a visual interpretation of the alignment between sequences. This makes it particularly useful for evaluating the consistency between predicted and actual time series trajectories, allowing for a more nuanced understanding of model behavior. Instead of discussing the DTW area as shown in the manuscript, this section focused on the full warping path, also presenting both predicted and expected values in the same plot.

In Figure 4, we illustrate the DTW warping paths for the Lorenz series, comparing the proposed ChaoticFuzz approach with LSTM. The warping path produced by ChaoticFuzz (Figure 4a) remains closer to the main diagonal (perfect match), indicating a more consistent temporal alignment with the reference series. In contrast, LSTM (Figure 4b) exhibits greater area between warping path and the diagonal, reflecting larger temporal distortions required to align its forecasts with the ground truth. These results suggest that ChaoticFuzz more effectively preserves the intrinsic temporal structure of the Lorenz system compared to LSTM.

Similar analyses were conducted on the remaining datasets. Figure 5 presents the DTW-based comparison between ChaoticFuzz and GRU for forecasting the Hénon time series. Consistent with previous results, ChaoticFuzz exhibits a smaller area between the warping path and the diagonal reference line, indicating a closer alignment with the ground truth and a better ability to capture the underlying dynamics.

Figure 6 illustrates the DTW-based comparison between ChaoticFuzz and GRU on the Rössler time series. As discussed earlier in the manuscript, this is one of the cases where our proposed method underperformed compared to the baseline. This observation is corroborated by the warping path in

the figure, which reveals a larger deviation from the diagonal, indicating a poorer alignment with the ground truth.

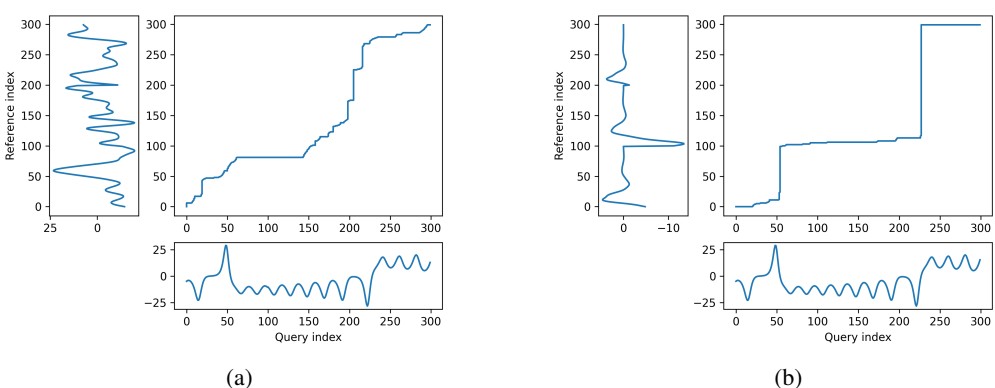

Figure 4: DTW warping path visualization for the Lorenz time series: (a) ChaoticFuzz; and (b) LSTM.

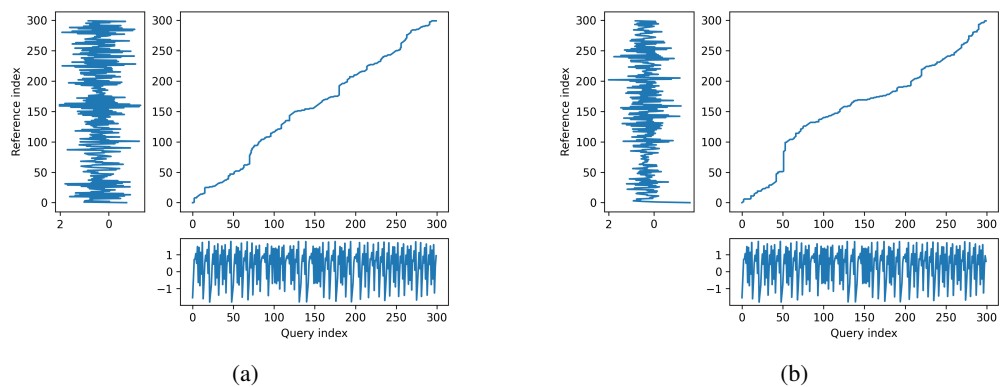

Figure 5: DTW warping path visualization for the Hénon series: (a) ChaoticFuzz; and (b) GRU.

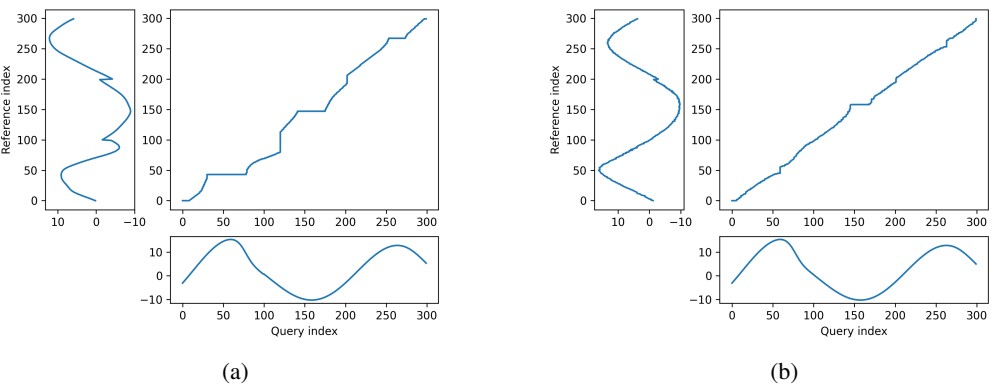

Figure 6: DTW warping path visualization for the Rössler series: (a) ChaoticFuzz (proposed); (b) GRU.

Figures 7 and 8 present the DTW-based visual analyses for the experiments conducted on the Logistic and Chua time series, respectively. In both cases, our proposed method outperforms the best-performing baselines—SARIMA for the Logistic series and GRU for the Chua series. The warping paths in both figures confirm that ChaoticFuzz consistently achieves superior temporal alignment with the ground truth, further highlighting its effectiveness in capturing the underlying dynamics of chaotic systems.

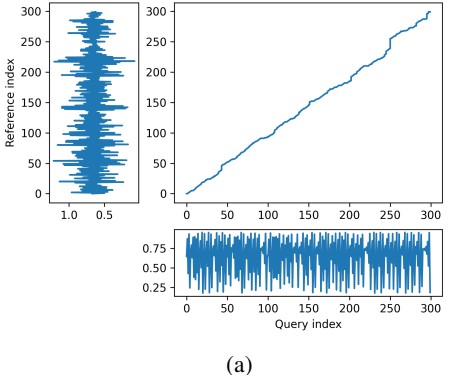 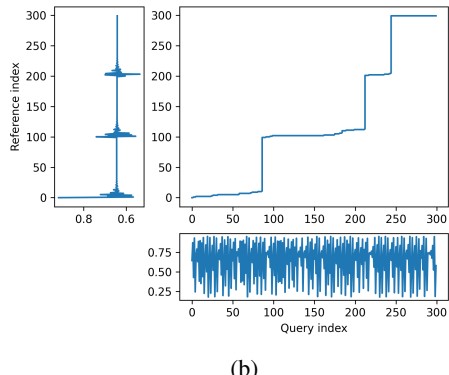

(a)          (b)

Figure 7: DTW warping path visualization for the Logistic series: (a) ChaoticFuzz; and (b) SARIMA.

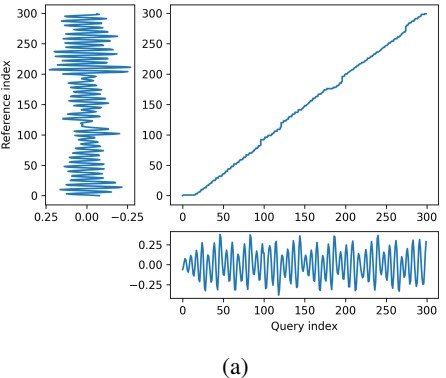 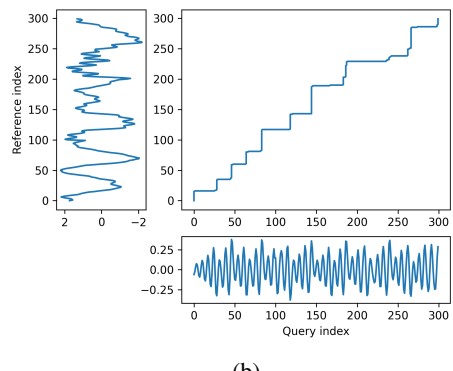

(a)          (b)

Figure 8: DTW warping path visualization for the Chua series: (a) ChaoticFuzz; and (b) GRU.

In summary, the visual analyses presented in this section align with the quantitative DTW measurements, further confirming that the proposed method consistently achieves closer temporal alignment than competing approaches across the evaluated chaotic time series.

In the following experiments, we evaluate the impact of the number of clusters used in constructing the graph structure within our ChaoticFuzz framework. While clustering is typically a critical step for pattern discovery and often sensitive to hyperparameters, particularly the number of clusters, our results indicate that ChaoticFuzz exhibits stable behavior across different cluster configurations. In other words, the number of clusters does not substantially affect the final prediction performance. To support this observation, Table 2 reports various regression metrics (DTW, MAE, MSE, RMSE, MAPE, and $R^2$) for the Lorenz time series. As in the main manuscript, our discussion primarily focuses on DTW and RMSE, given their greater relevance for evaluating long-term forecasting and dynamic alignment in chaotic time series.

We replicated the same experimental setup for the remaining datasets. As shown in Tables 3, 4, 5, and 6, the predictive performance of our approach remains consistent across different numbers of clusters. This robustness is largely attributed to the fuzzy membership functions, which provide

Table 2: Performance metrics by group - Lorenz system.

| Group | DTW | MAE | MSE | RMSE | MAPE | $R^2$ |
|------:|-----|-----|-----|------|------|-------|
| 3 | 2.34 | 10.23 | 174.98 | 13.23 | 10.84 | -0.22 |
| 6 | 3.35 | 11.92 | 250.99 | 15.84 | 8.60 | -0.75 |
| 9 | 3.09 | 12.31 | 233.45 | 15.28 | 6.15 | -0.63 |
| 12 | 2.51 | 12.06 | 232.81 | 15.26 | 4.67 | -0.62 |
| 24 | 3.20 | 12.27 | 225.79 | 15.03 | 2.11 | -0.57 |

a soft association between phase-space observations and clusters, allowing the model to capture nuanced relationships even as the number of clusters varies. To maintain consistency and simplify our analysis, we fixed the number of clusters to 12 across all experiments.

Table 3: Performance metrics by group - Henon map.

| Group | DTW | MAE | MSE | RMSE | MAPE | $R^2$ |
|------:|-----|-----|-----|------|------|-------|
| 3 | 0.42 | 1.04 | 1.61 | 1.27 | 2.10 | -0.52 |
| 6 | 0.48 | 1.25 | 2.29 | 1.51 | 2.44 | -1.16 |
| 9 | 0.41 | 0.99 | 1.47 | 1.21 | 1.91 | -0.39 |
| 12 | 0.40 | 1.04 | 1.62 | 1.27 | 2.01 | -0.54 |
| 24 | 0.42 | 0.99 | 1.40 | 1.18 | 1.64 | -0.32 |

Table 4: Performance metrics by group - Rössler map.

| Group | DTW | MAE | MSE | RMSE | MAPE | $R^2$ |
|------:|-----|-----|-----|------|------|-------|
| 3 | 0.22 | 1.23 | 2.37 | 1.54 | 0.42 | 0.97 |
| 6 | 0.27 | 1.79 | 5.57 | 2.36 | 0.46 | 0.92 |
| 9 | 0.54 | 1.86 | 6.38 | 2.53 | 0.41 | 0.91 |
| 12 | 0.55 | 3.04 | 21.69 | 4.66 | 1.01 | 0.69 |
| 24 | 0.25 | 1.86 | 5.49 | 2.34 | 0.45 | 0.92 |

Table 5: Performance metrics by group - Logistic map.

| Group | DTW | MAE | MSE | RMSE | MAPE | $R^2$ |
|------:|-----|-----|-----|------|------|-------|
| 3 | 0.11 | 0.30 | 0.14 | 0.37 | 0.70 | -1.21 |
| 6 | 0.11 | 0.27 | 0.12 | 0.35 | 0.62 | -1.01 |
| 9 | 0.10 | 0.26 | 0.11 | 0.33 | 0.63 | -0.74 |
| 12 | 0.10 | 0.28 | 0.12 | 0.35 | 0.67 | -0.94 |
| 24 | 0.10 | 0.25 | 0.09 | 0.30 | 0.59 | -0.45 |

Table 6: Performance metrics by group - Chua's Circuit.

| Group | DTW | MAE | MSE | RMSE | MAPE | $R^2$ |
|------:|-----|-----|-----|------|------|-------|
| 3 | 0.07 | 0.15 | 0.03 | 0.18 | 1.62 | 0.01 |
| 6 | 0.08 | 0.15 | 0.03 | 0.18 | 1.19 | -0.05 |
| 9 | 0.07 | 0.17 | 0.04 | 0.20 | 1.54 | -0.25 |
| 12 | 0.06 | 0.17 | 0.04 | 0.21 | 1.65 | -0.34 |
| 24 | 0.06 | 0.16 | 0.04 | 0.19 | 4.88 | -0.09 |

Finally, although our primary focus is on forecasting time-domain observations, Figure 9 illustrates the behavior of both expected and predicted observations in the phase space for the Lorenz system. As shown, the predictions generated by ChaoticFuzz closely follow the structure of the true attractor, preserving the overall geometry and dynamic behavior of the system. This reinforces the effectiveness of our spatiotemporal modeling approach in capturing not only the temporal evolution but also the underlying system dynamics.

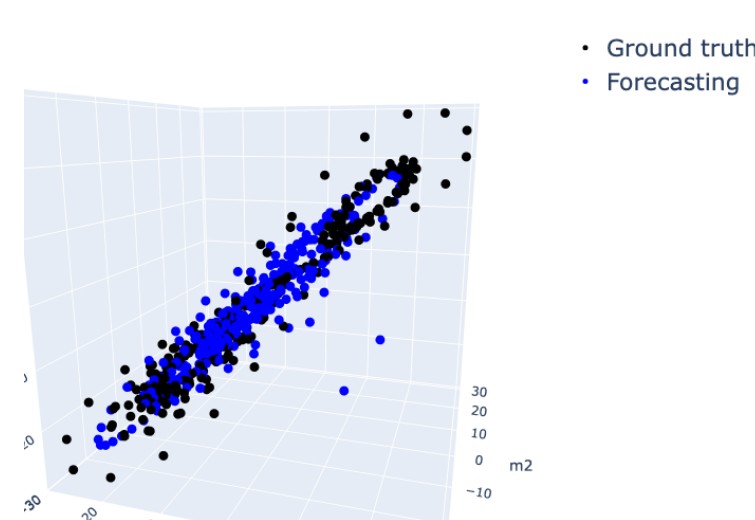

Figure 9: Expected and predicted observations in phase space - Lorenz system.

## C   HYPERPARAMETERS

For the clustering stage, we applied the Fuzzy C-Means algorithm (R package 'ppclust') on the two-dimensional phase-space reconstruction. The model was configured with 12 clusters to all datasets, using the default fuzzification parameter m = 1.5, Euclidean distance, and a convergence criterion based on the reduction of the objective function up to the internal tolerance (eps = 1e-06). The algorithm ran up to 1000 iterations to ensure the stability of the fuzzy memberships.

All models shared the same input-output configuration and training protocol to ensure a fair comparison. Specifically, the prediction horizon was fixed to 100 future time steps, the sliding window size (input sequence length) was set to 100, and each sample contained a single input feature. Training was conducted for 1000 epochs using mini-batches of size 32, the Adam optimizer (learning rate = 0.001), and mean squared error (MSE) loss.

For the recurrent baselines (GRU and LSTM), both models consist of two stacked recurrent layers with 64 hidden units each. A fully connected layer is applied to the last hidden state of the encoder to output the multi-step forecast of 100 steps. No dropout or layer normalization was applied to keep the architecture minimal.

The foundation models CHRONOS and TimesFM (200M and 500M parameters) were not fine-tuned. We used their publicly available pretrained checkpoints in a zero-shot forecasting setting, following the official inference procedures.

SARIMA parameters $(p, d, q)$ and seasonal $(P, D, Q, s)$ were automatically selected using the `pmdarima` library's `auto_arima` function, searching over $p, q \in [0, 3]$, $d \in [0, 2]$, and a seasonal period $s = 30$, optimizing both AIC and BIC scores. Figure 7 shows the fit parameters for each dataset.

For the graph-based models (GCN, GraphSAGE, Cheb, GAT, and LEConv) all architectures share a consistent design. Each model employs a single graph convolutional layer with 64 hidden units and ReLU activation. The resulting node embeddings are aggregated using global attention and subsequently passed through a fully connected decoder to predict two target variables over a 100-step horizon. Each input graph consists of 12 nodes, each represented by 100-dimensional feature vectors.

For the models Attraos, DLinear, and PatchTST, we report both the common training configuration applied across experiments as well as the model-specific parameters for each architecture. By explicitly reporting these values, we aim to facilitate fair comparison between baselines and enable replication of our results.

Table 7: SARIMA Parameters and Hyperparameters after Grid Search

| Series | $p$ | $d$ | $q$ | $P$ | $D$ | $Q$ | $s$ |
|--------|-----|-----|-----|-----|-----|-----|-----|
| Lorenz | 4 | 0 | 4 | 0 | 0 | 0 | 30 |
| Hénon | 5 | 0 | 2 | 0 | 0 | 0 | 30 |
| Rössler | 0 | 0 | 0 | 2 | 0 | 1 | 30 |
| Logistic | 5 | 0 | 5 | 0 | 0 | 0 | 30 |

**Task and Data.** All experiments followed the long-term forecasting protocol on a custom dataset, in a univariate setting (`features=S`, `target=ts`). Temporal features used hourly encoding (`freq=h`), and seasonality metadata was set to `Monthly`.

**Windowing and Horizon.** We used an input sliding window of `seq_len=` 100 and a fixed prediction horizon of `pred_len=` 100 (with `label_len=` 0). Unless otherwise stated by the data adapter, input/output channel sizes followed repository defaults.

**Optimization.** Models were trained for `train_epochs=` 100 with mini-batches of `batch_size=` 32, Adam optimizer with learning rate $1 \times 10^{-4}$ and MSE loss. Early stopping patience was `patience=` 3. Mixed precision was disabled (`use_amp=`False). We performed one run per configuration (`itr=` 1). DTW-based metrics and all data augmentations were disabled.

MODEL-SPECIFIC SETTINGS

**Attraos.** Phase-space reconstruction was enabled with `PSR_dim=` 4, `PSR_delay=` 12, independent channels (`PSR_type=`indep), hierarchical projection on (`multi_res=`True), and FFT-based evolution off (`FFT_evolve=`False). The backbone used a transformer-style configuration with `d_model=` 512, `n_heads=` 8, `e_layers=` 2, `d_layers=` 1, and `d_ff=` 2048, time features embedding (`embed=`timeF), GELU activation, dropout = 0.1, and encoder distillation enabled (`distil=`True). RevIN was active (`revin=`1, `affine=`0, `subtract_last=`0).

**DLinear.** We followed the repository defaults for DLinear. Since DLinear is a linear forecaster, transformer-specific knobs (e.g., `n_heads`, `e_layers`, `d_layers`, `d_ff`, `embed`) are not used by the model implementation. Likewise, Attraos PSR flags (`PSR_*`) are not consumed by DLinear. Effective training setup (window, horizon, optimizer, epochs, batch size, loss, patience) matches the common configuration above.

**PatchTST.** We used the common input/output setup and optimizer. PatchTST-specific patching parameters kept repository defaults: `patch_size=` 16 and `patch_stride=` 8 (`stride` in the parser). The transformer backbone followed `d_model=` 512, `n_heads=` 8, `e_layers=` 2, `d_layers=` 1, `d_ff=` 2048, `embed=`timeF, GELU, dropout = 0.1, with encoder distillation on (`distil=`True). Attraos PSR flags (`PSR_*`) are not used by PatchTST.

REPRODUCIBILITY (CLI)

**Attraos:**

```
python run.py --task_name long_term_forecast --is_training 1 \
  --model Attraos --data custom --root_path ./dataset/ \
  --data_path lor64.csv --features S --target ts --freq h \
  --seq_len 100 --label_len 0 --pred_len 100 \
  --d_model 512 --n_heads 8 --e_layers 2 --d_layers 1 \
  --d_ff 2048 --batch_size 32 --train_epochs 100 \
  --learning_rate 1e-4 --seasonal_patterns Monthly --PSR_dim 4 \
  --PSR_delay 12 --PSR_type indep --multi_res True \
  --FFT_evolve False
```

**DLinear:**

```
python run.py --task_name long_term_forecast \
  --is_training 1 --model DLinear --data custom \
  --root_path ./dataset/ --data_path lor64.csv --features S \
  --target ts --freq h --seq_len 100 --label_len 0 --pred_len 100 \
  --d_model 512 --n_heads 8 --e_layers 2 --d_layers 1 --d_ff 2048 \
  --batch_size 32 --train_epochs 100 \
  --learning_rate 1e-4 --seasonal_patterns Monthly
```

**PatchTST:**

```
python run.py --task_name long_term_forecast --is_training 1 \
  --model PatchTST --data custom --root_path ./dataset/ \
  --data_path lor64.csv --features S --target ts --freq h \
  --seq_len 100 --label_len 0 --pred_len 100 --d_model 512 \
  --n_heads 8 --e_layers 2 --d_layers 1 --d_ff 2048 \
  --batch_size 32 --train_epochs 100 --learning_rate 1e-4 \
  --seasonal_patterns Monthly
```

## D  COMPUTATIONAL SETUP

All experiments were conducted on a machine with the following configuration:

- **CPU:** 2× Intel Xeon Gold 6326 @ 2.90GHz (32 physical cores, 64 threads total)
- **GPU:** 4× NVIDIA A16 (16GiB each)
- **RAM:** 125GiB
- **Swap Memory:** 8GiB
- **Operating System:** Ubuntu 22.04 with kernel 5.15.0-144-generic
- **Virtualization:** VT-x supported

Detailed cache configuration includes L1d: 1.5MiB, L1i: 1MiB, L2: 40MiB, and L3: 48MiB. The system was NUMA, enabled with two memory nodes.

