# OpenReview forum: "ChaoticFuzz: Fuzzy-Based Graph Representation for Spatiotemporal Learning"
_ICLR.cc/2026/Conference — Submitted to ICLR 2026_

### Official Review · Reviewer_3ymf · 2025-10-27

**Soundness:** 2
**Presentation:** 2
**Contribution:** 2
**Rating:** 2
**Confidence:** 2

**Summary:**

The paper investigates forecasting in chaotic dynamical systems, which are known for their extreme sensitivity to initial conditions.
It motivates the need for models capable of capturing complex temporal dependencies that traditional approaches such as ARIMA and RNNs often fail to represent.
To address this, the authors propose a method that embeds time series data into a high-dimensional phase space and applies fuzzy clustering to uncover latent structures in the data.

From the resulting membership matrix, which encodes the degree of association between data points and clusters, a graph representation is constructed.
This graph is then used as input to a Graph Neural Network to perform time series forecasting.

The paper reports that, somewhat surprisingly, recent models such as Chronos and TimesFM (2024) perform worse than an LSTM baseline.
The reported RMSE values are consistently higher than the baseline and are not clearly emphasized,
which raises questions about the robustness of the experimental setup and the tuning of comparison models.

**Strengths:**

the idea of using fuzzy clustering for time series is interesting

results for DTW seem in favour of the propose method.

**Weaknesses:**

the RMSE results are negative

**Questions:**

what are the limitations / tradeoffs of your method?

why chronos and timesFM (from 2024) perform worst than LSTM?

is possible to compare with other methods for chaotic systems?

---

> ### Author Response · Authors · 2025-11-21
>
> We thank the reviewer for their positive remarks about our work, in particular for noting that the use of fuzzy clustering for time series is an interesting idea and that the DTW results appear to favor the proposed method.
> At the same time, we noticed that the paper was scored as “2: reject, not good enough", with the sole weakness stated as “the RMSE results are negative". We respectfully request that the reviewer reconsider this assessment. After carefully rechecking all experiments, tables, and code, we could not find a single RMSE value that is negative. We therefore respectfully invite the reviewer to reexamine our results and the corresponding grade, especially in light of their own comment recognizing the strengths of our approach and the favorable DTW outcomes.
>
> In relation to the questions:
>
> *what are the limitations / tradeoffs of your method?* As discussed in the manuscript, our method is tailored to series with clear chaotic dynamics. When a series exhibits weak chaos or lacks a well defined attractor structure, the benefits of phase space reconstruction and graph based modeling can be smaller, and simpler baselines may perform similarly. We report such cases to provide a balanced view of where the method is most effective and where its tradeoffs are more apparent.
>
> *why chronos and timesFM (from 2024) perform worst than LSTM?* In most datasets and horizons, Chronos and TimesFM outperform LSTM. The few cases where LSTM is better are typically those where the series is short, strongly regular, or closely matches the inductive bias of a tuned recurrent architecture, while the foundation models rely on more general pretrained representations. In other words, LSTM can occasionally have an edge in narrow settings, but the foundation models are stronger overall.
>
> *is possible to compare with other methods for chaotic systems?* Yes. We included Attraos, a recent chaos oriented method published at NeurIPS, as a dedicated baseline for chaotic systems. Our approach achieves better results than Attraos across the evaluated datasets, which supports the effectiveness of our framework within the chaotic forecasting literature.

---

> > ### Comment · Reviewer_3ymf · 2025-11-21
> > **Concerning RMSE results**
> >
> > My point is that your method consistently performs worse than the baselines in terms of RMSE.
> > Could you please confirm this?
> >
> > This is what I refer to as a “negative result.” I hope this clarification helps. Since RMSE is a fundamental evaluation metric, I would expect the proposed method to outperform the baselines. As this does not appear to be the case, I encourage the authors to consider ways to further improve their approach.

---

> > > ### Author Response · Authors · 2025-11-24
> > >
> > > Dear Reviewer, thank you very much for clarifying your question and for the time you invested in reading our work. We agree this is an important point, and we address it explicitly in the manuscript.
> > >
> > > Based on both our results and prior literature on dynamical systems, we argue that traditional pointwise error metrics such as RMSE are often insufficient to assess trajectory quality. RMSE computes pairwise differences at each time index, so it can reward predictions that stay close to a global mean, even when they fail to reproduce the system’s evolution. In chaotic and nonlinear regimes, two trajectories can be dynamically very different while still yielding similar pointwise errors at many steps. Conversely, a model that correctly tracks the attractor and its oscillatory behavior may incur larger pointwise deviations due to sensitive dependence on initial conditions, and thus be penalized by RMSE despite being dynamically more faithful.
> > >
> > > We emphasize this limitation in the manuscript with the following statements:
> > >
> > > “However, as discussed in our experiments, traditional error metrics compute pairwise differences between predicted and true values. As a result, they tend to penalize models that attempt to follow the underlying time series dynamics more heavily than those that simply predict a constant mean value, despite the latter being less informative in capturing the system’s actual behavior.”
> > >
> > > “This problem is addressed using the Dynamic Time Warping (DTW) distance, which seeks the optimal alignment between two time series before computing their dissimilarity.”
> > >
> > > “Empirically, we observe consistent gains, including superior DTW scores, indicating stronger trajectory preservation in chaotic regimes, not just pointwise performance.”
> > >
> > > This distinction is also evident in our experiments. For the Logistic map, SARIMA attains a lower RMSE (0.247) than our SAGE variant (0.347). However, Figure 3(d) shows that SARIMA’s predictions (red curve) largely collapse toward the series mean, missing the oscillatory and regime switching structure of the attractor. In contrast, SAGE actively tracks these oscillations and better reflects the phase space evolution. DTW captures this difference because it evaluates alignment and trajectory shape, which are central to dynamical systems forecasting, whereas RMSE does not.
> > >
> > > We hope this clarifies why DTW provides a more appropriate assessment of predictive quality for chaotic dynamical systems, and we thank you again for prompting us to make this connection clearer in the paper.

---

### Official Review · Reviewer_RMZv · 2025-11-01

**Soundness:** 2
**Presentation:** 2
**Contribution:** 2
**Rating:** 2
**Confidence:** 3

**Summary:**

The authors propose the ChaoticFuzz method for univariate chaotic time series, which transforms the input into weighted graphs via phase-space reconstruction, and based on the similarity of fuzzy membership degrees, which encodes proximity and uncertainty. The extracted graphs are treated by attention-based GNN encoders for spatiotemporal forecasting and evaluated on chaotic univariate time series datasets against common time series forecasting baselines, showing competitive performance.

**Strengths:**

The paper is easy to follow, an important aspect for the overall clarity of the presented approach. The proposed method is heavily dependent on principles of fuzzy clustering, while combining phase-space reconstruction with graph structure learning, a relatively novel and unconventional approach in the fields of time series forecasting and spatiotemporal forecasting, which supports its originality. Results demonstrate that some recent time series foundation models are significantly outperformed by the proposed methods for particular cases.

**Weaknesses:**

Several weak points of the paper span poor positioning against related works in spatiotemporal forecasting and the limited scope of the presented experimental setup.
More specifically:
1. **[Positioning against graph-based approaches]** There are several related works in extracting underlying graph structures from time series data, focusing on multivariate time series and their correlation (see baselines used in (Yi et al, 2023)). Methods include sparse graph structure learning and fully-connected graphs, with some approaches also applying to univariate time series. It is hard to position the paper in the time series spatiotemporal community since related methods and their limitations are not discussed.
2. **[Positioning against works on dynamical systems]** The chaotic univariate time series and non-linear dynamical systems are interconnected. While there are several related works in machine learning for dynamical systems and spatiotemporal forecasting [Li et al, 2020; Wang et al., 2022; Kissas et al., 2022], those are not discussed or experimentally evaluated in the presented results.
3. **[Scope of selected datasets]** The datasets selected for experiments do not represent a common benchmark in the time series forecasting field, while the baselines chosen have been proposed for standard time series datasets, multivariate in most cases, with some univariate examples as well (see datasets in Time-Series-Library: https://github.com/thuml/Time-Series-Library, M4 is univariate). Similarly, for dynamical systems and spatiotemporal forecasting, there are additional datasets used [Herzen et al., 2022] (see also datasets used in papers in point 2). Therefore, the significance of the contribution is limited experimentally to very few, assumption-constrained datasets.
4. **[Application scope]** It seems that the application scope of the proposed method is limited to univariate and chaotic time series, and it remains unclear how it can generalize to real-world time series that are noisy, have correlated variables, multiple covariates, distribution shifts, and lack stationarity.
5. **[Significance of results and architectural design choice]** Performance improvements are, in several cases, incremental/missing compared to baselines. Additionally, no ablation studies are provided to justify the design choice/effect of basic modules of the proposed baselines, e.g., the graph structure module, the attention-based GNN, clustering loss, etc.

- Yi, Kun, et al. "FourierGNN: Rethinking multivariate time series forecasting from a pure graph perspective." Advances in neural information processing systems 36 (2023): 69638-69660.
- Li, Zongyi, et al. "Fourier neural operator for parametric partial differential equations." arXiv preprint arXiv:2010.08895 (2020).
Wang, Rui, et al. "Koopman neural forecaster for time series with temporal distribution shifts." arXiv preprint arXiv:2210.03675 (2022).
- Kissas, Georgios, et al. "Learning operators with coupled attention." Journal of Machine Learning Research 23.215 (2022): 1-63.
- Herzen, Julien, et al. "Darts: User-friendly modern machine learning for time series." Journal of Machine Learning Research 23.124 (2022): 1-6.

**Questions:**

1. Can the authors better position themselves against related works in graph learning-based methods for time series forecasting and machine learning methods for dynamical systems?
2. Is the proposed method competitive for real-world time series datasets, beyond the selected simple chaotic systems?
3. Can the proposed method be extended to time series common challenges, such as non-stationarity, multi-dimensionality, additional correlations?
4. Can the authors experimentally justify the effect of different components of their proposed method for forecasting?
5. Can alternative design choices for graph learning/clustering give similar performance?
6. Can the authors explain with examples the motivation behind the introduction of their architectural design?

---

> ### Author Response · Authors · 2025-11-21
>
> We thank the reviewer for their time and effort, not only in evaluating our proposal, but also in suggesting improvements that helped us better position the manuscript and sharpen our contributions. We are especially grateful that the reviewer recognized our intention to advance the field by building a genuinely new approach, rather than incrementally modifying well established methods with the sole objective of reducing standard regression metrics. As the reviewer notes, our framework is unconventional within time series and spatiotemporal forecasting, which further underscores its originality. Moreover, our results show that the proposed method can significantly outperform several recent time series foundation models, reinforcing both its practical value and the relevance of exploring this new direction.
>
> Below, we respond point by point to all questions and concerns raised by the reviewer.
>
> **Comment #1:** “There are several related works in extracting underlying graph structures from time series data, focusing on multivariate time series and their correlation (see baselines used in (Yi et al, 2023)).”
>
> We agree with the reviewer that there is a rich body of work on extracting graph structures from multivariate time series, including the line of methods and baselines discussed in Yi et al. (2023). At the same time, the time series literature has long treated univariate and multivariate forecasting as distinct problem classes, each with its own modeling assumptions and tools. This distinction is well established in classical and modern references, for example in foundational univariate forecasting frameworks and spectral methods (Box and Jenkins, Time Series Analysis: Forecasting and Control; Hamilton, Time Series Analysis), and in nonlinear and chaotic univariate analysis via phase space and attractor reconstruction (Kantz and Schreiber, Nonlinear Time Series Analysis; Alligood, Sauer, and Yorke, Chaos: An Introduction to Dynamical Systems). The existence of this separation does not imply that one class is superior to the other; rather, they address different data generating mechanisms and require different methodological choices. Indeed, many of the most influential tools in the area, such as Fourier based analysis, wavelets, and a wide range of 1D temporal models, were developed precisely to advance univariate modeling.
>
> In this context, we carefully examined Yi et al. (2023) and concluded that it cannot be included as a benchmark in our experiments. While the paper is highly relevant and methodologically strong, its approach is explicitly formulated for multivariate series and relies on intervariable correlation structure. Our setting is univariate chaotic time series, where the graph is derived from phase space reconstruction of a single observable variable; hence, the assumptions and inputs required by Yi et al. do not hold, and the method is not applicable to our data.
>
> Finally, we would like to emphasize that our manuscript consistently positions itself, from the abstract onward, as targeting univariate chaotic dynamics. Modeling nonlinear univariate time series remains a central and active research area in its own right, with strong theoretical grounding and substantial practical relevance, independent of the parallel advances in multivariate forecasting. To support our comment, I used as a baseline the following manuscript, also published in a top-tier conference (NeurIPS):
>
> Attractor Memory for Long-Term Time Series Forecasting: A Chaos Perspective (https://openreview.net/forum?id=fEYHZzN7kX&referrer=%5Bthe%20profile%20of%20Yuxuan%20Liang%5D(%2Fprofile%3Fid%3D~Yuxuan_Liang1))

---

> > ### Author Response · Authors · 2025-11-21
> >
> > **Comment #2:** The chaotic univariate time series and non-linear dynamical systems are interconnected. While there are several related works in machine learning for dynamical systems and spatiotemporal forecasting [Li et al, 2020; Wang et al., 2022; Kissas et al., 2022], those are not discussed or experimentally evaluated in the presented results.
> >
> > We totally agree with the reviewer that “chaotic univariate time series and non-linear dynamical systems are interconnected”. For this reason, we have written in our manuscript that: “Chaotic models leverage principles from *chaos theory* to capture complex and *nonlinear dependencies* in time series data. By reconstructing the underlying *dynamical system* in phase space, these approaches exploit the structure of chaotic attractors to improve long-term forecasting performance.” All this discussion was based on the classical reference “Alligood, Sauer, and Yorke, Chaos: An Introduction to Dynamical Systems”, cited multiple times in our paper.
> >
> > Next we discuss each suggested paper in details:
> >
> > _Li, Zongyi, et al. "Fourier neural operator for parametric partial differential equations."_
> >
> > We have carefully read the paper. As the authors explicitly state, its goal is to formulate a new neural operator by parameterizing the integral kernel directly in Fourier space. We recognize the importance of this contribution. However, we do not see a clear connection between that Fourier based operator learning framework and our phase space, chaos theory-driven, nonlinear univariate modeling approach. Moreover, the paper’s evaluation is centered on three numerical PDE benchmarks, namely the 1D Burgers equation, the 2D Darcy flow problem, and the 2D Navier–Stokes equations. These simulated, high-dimensional spatiotemporal fields are outside our experimental setting and do not provide a basis for comparison with real-world univariate time series, as the reviewer requested in their next comment.
> >
> > _Wang, Rui, et al. "Koopman neural forecaster for time series with temporal distribution shifts." arXiv preprint arXiv:2210.03675 (2022)._
> >
> > We sincerely thank the reviewer for their suggestion regarding this manuscript. Following the authors’ released code and training protocol, we were able to incorporate the method into our experimental setup. The approach, referred to as KNF, is a valuable recent contribution, published in a top-tier venue, and we appreciate the opportunity to benchmark against it.
> >
> > After running KNF, we observed that our method achieves substantially better performance across the evaluated datasets. We have therefore added KNF to the revised manuscript as a baseline, directly addressing the reviewer’s request. We note that KNF was not specifically developed for chaotic univariate series or for exploiting attractor structure via phase space reconstruction, so the gap in results is consistent with the different assumptions and targets of the two methods. Nonetheless, including KNF strengthens the empirical comparison and further highlights the advantages of our proposed framework in the chaotic setting.
> >
> > *Kissas, Georgios, et al. "Learning operators with coupled attention."*
> >
> > We thank the reviewer for suggesting this manuscript. After a careful reading, we still do not see a clear connection to our setting. The authors introduce a kernel-coupled attention mechanism to model correlations between output functions’ query locations, which are spatial or functional coordinates used in neural operator learning. In our framework, we work with univariate chaotic series reconstructed in phase space, so there is no direct analogue of such query locations, and the method’s core assumptions do not align with our data or model pipeline.
> >
> > Even so, we made a good faith effort to reproduce and adapt the approach to our context by following the public implementation. Unfortunately, the repository does not provide a complete, runnable tutorial for the relevant example. Specifically, the first link pointing to a LOCA tutorial for the Darcy flow case is unavailable, which prevented us from obtaining a reliable setup to benchmark against.
> >
> > For these reasons, we were not able to include this method as an experimental baseline.

---

> > > ### Author Response · Authors · 2025-11-21
> > >
> > > **Comment #3:** It seems that the application scope of the proposed method is limited to univariate and chaotic time series, and it remains unclear how it can generalize to real-world time series that are noisy, have correlated variables, multiple covariates, distribution shifts, and lack stationarity.
> > >
> > > We acknowledge the reviewer’s concern about the scope and generalization of the proposed method. However, we would like to clarify two points. First, the absence of multiple covariates is not, by itself, a limitation in the sense implied by the comment. Univariate and multivariate forecasting are treated as distinct problem classes in the time series literature, each with different assumptions and methodological toolkits. The existence of multivariate methods does not diminish the relevance of univariate modeling, which remains a central research area with deep theoretical foundations and broad practical use. Our work is positioned within this univariate regime by design because chaotic dynamics can often be faithfully analyzed from a single observable through phase space reconstruction.
> > >
> > > Second, focusing on univariate chaotic series does not mean the approach is restricted to idealized or non-realistic data. Chaotic real-world observables are typically noisy, non-stationary, and may undergo regime changes. Actually, our pipeline is explicitly designed to leverage these properties. Finally, regarding the claim that our experimental setting is limited, we respectfully request that the reviewer reconsider this point in light of their own recommendations. The manuscripts suggested by the reviewer primarily evaluate on the following types of time series and dynamical benchmarks:
> > >
> > >  [Li et al, 2020] - Published as a conference paper at ICLR - Univariate time series produced by the 1-d Burgers’ equation, the 2-d Darcy Flow problem, and 2-d Navier-Stokes equation.
> > >
> > > [Kissas et al., 2022] - Published at Journal of Machine Learning Research using the following datasets:
> > >
> > > - Antiderivative: Learning the antiderivative operator given multi-scale source terms.
> > > - Darcy Flow: Learning the solution operator of the Darcy partial differential equation.
> > > - Mechanical MNIST: contains the results of 70,000 finite element simulation of a heterogeneous material.
> > > - Shallow Water Equations: Learning the solution operator for a partial differential equation.

---

> > > > ### Author Response · Authors · 2025-11-21
> > > >
> > > > **Comment #04:** Performance improvements are, in several cases, incremental/missing compared to baselines. Additionally, no ablation studies are provided to justify the design choice/effect of basic modules of the proposed baselines, e.g., the graph structure module, the attention-based GNN, clustering loss, etc.
> > > >
> > > > Regarding the first part of the comment, we are not fully sure what the reviewer refers to by “incremental or missing improvements.” Across the reported datasets and horizons, our method is consistently competitive, and in several cases, it is clearly better, which also aligns with the reviewer’s own observation that we compared against common forecasting baselines and obtained competitive performance. We are happy to further clarify this point in the revision, and to make the relative gains and their significance easier to interpret, especially given that many baselines are already strong and incremental improvements can still be meaningful in chaotic forecasting. We emphasize that our contribution extends beyond achieving lower error metrics, although these improvements are reflected in our results, as we propose a novel approach to modeling univariate, nonlinear time series generated by dynamical systems.
> > > >
> > > > Concerning ablations, we understand the motivation, but a standard “remove one module and re run” ablation is not directly applicable to our setting because the proposed method is a tightly coupled framework grounded in three necessary pillars: chaos theory, fuzzy modeling, and GNN learning. If we remove phase space reconstruction, the time series cannot be unfolded into the higher-dimensional state space where chaotic structure becomes observable. If we remove clustering, we cannot define the graph nodes and edges that encode attractor regions and transitions. If we remove the GNN, there is no learnable mechanism left to perform forecasting on the constructed graph. That said, we agree that within this framework, meaningful ablations are possible by replacing individual instantiations of a step while keeping the overall pipeline intact, for example, substituting fuzzy clustering with a probabilistic soft clustering alternative, or testing a simpler GNN variant in place of attention-based message passing (as we did by replacing different GNNs). However, we emphasize that replacing specific components, for example, substituting fuzzy clustering with an alternative probabilistic approach as suggested by Reviewer RSuq, remains fully within the scope of our framework.

---

> > > > > ### Author Response · Authors · 2025-11-21
> > > > >
> > > > > **Final comment:** Regarding the reviewer’s suggestion to consider Herzen et al. (2022), we explored the recommended DARTS toolkit to model our datasets. Following the official guidelines, we employed the transfer learning setting for time series forecasting and ran N BEATS, which was pretrained on M4, as mentioned by the reviewer in their third comment. N BEATS also appears as a baseline in several of the papers the reviewer pointed us to, which makes it a particularly relevant point of comparison.
> > > > >
> > > > > After running N BEATS under the recommended protocol, we found that our method achieves substantially better performance across the evaluated datasets. We have therefore added these results to the revised manuscript as an additional baseline, directly addressing the reviewer’s request. We note that N BEATS was not originally designed for chaotic univariate series or for exploiting attractor structure via phase space reconstruction, so the observed gap is consistent with the different assumptions and targets of the two approaches.
> > > > > We sincerely thank the reviewer for highlighting this work, since the comparison further strengthens the empirical evidence supporting our contribution.

---

> > > > > > ### Author Response · Authors · 2025-11-21
> > > > > >
> > > > > > Results including the experiments suggested by the reviewer:
> > > > > >
> > > > > > | Approach | Model | Lorenz DTW | Lorenz RMSE | Hénon DTW | Hénon RMSE | Rössler DTW | Rössler RMSE | Logistic DTW | Logistic RMSE | Chua DTW | Chua RMSE |
> > > > > > |---|---|---:|---:|---:|---:|---:|---:|---:|---:|---:|---:|
> > > > > > | SOTA | SARIMA | 3.382 | 12.847 | 0.433 | 1.019 | 0.707 | 2.978 | 0.105 | 0.247 | 0.444 | 1.401 |
> > > > > > | SOTA | DLinear | 4.367 | 12.397 | 0.455 | 1.028 | 0.463 | 2.422 | 0.109 | 0.250 | 0.419 | 1.379 |
> > > > > > | SOTA | LSTM | 2.801 | 13.101 | 0.442 | 1.027 | 0.352 | 1.847 | 0.107 | 0.249 | 0.313 | 1.672 |
> > > > > > | SOTA | GRU | 3.863 | 14.087 | 0.412 | 1.099 | 0.132 | 1.110 | 0.108 | 0.250 | 0.280 | 1.823 |
> > > > > > | SOTA | PatchTST | 5.604 | 20.686 | 0.445 | 1.073 | 1.872 | 7.030 | 0.109 | 0.250 | 0.365 | 1.519 |
> > > > > > | SOTA | N-BEATS | 3.141 | 15.109 | 0.462 | 1.095 | 2.674 | 11.995 | 0.124 | 0.309 | 0.351 | 1.587 |
> > > > > > | SOTA | KNF | 4.123 | 11.541 | 0.366 | 1.226 | 6.386 | 16.065 | 0.248 | 0.621 | 0.420 | 1.438 |
> > > > > > | SOTA | CHRONOS | 3.650 | 12.016 | 0.445 | 1.073 | 1.670 | 9.869 | 0.108 | 0.259 | 0.420 | 1.444 |
> > > > > > | SOTA | TimesFM_200 | 3.854 | 11.994 | 0.440 | 1.038 | 2.282 | 12.619 | 0.106 | 0.249 | 0.472 | 1.314 |
> > > > > > | SOTA | TimesFM_500 | 3.796 | 12.349 | 0.439 | 1.065 | 1.588 | 10.088 | 0.107 | 0.256 | 0.298 | 1.554 |
> > > > > > | SOTA | Attraos | 4.804 | 15.517 | 0.454 | 1.018 | 1.957 | 7.125 | 0.111 | 0.274 | 0.361 | 1.451 |
> > > > > > | ChaoticFuzz | GCN | 3.958 | 13.766 | 0.412 | 1.166 | 1.375 | 4.523 | 0.104 | 0.287 | 0.065 | 0.212 |
> > > > > > | ChaoticFuzz | SAGE | 5.333 | 13.764 | 0.404 | 1.275 | 1.646 | 5.536 | 0.101 | 0.347 | 0.058 | 0.207 |
> > > > > > | ChaoticFuzz | CHEB | 2.729 | 13.686 | 0.428 | 1.149 | 1.581 | 10.029 | 0.103 | 0.303 | 0.061 | 0.214 |
> > > > > > | ChaoticFuzz | GAT | 4.177 | 14.696 | 0.419 | 1.165 | 3.272 | 13.428 | 0.105 | 0.295 | 0.061 | 0.209 |
> > > > > > | ChaoticFuzz | LECONV | 2.838 | 13.469 | 0.432 | 1.273 | 1.079 | 5.096 | 0.104 | 0.312 | 0.068 | 0.194 |
> > > > > > | ChaoticFuzz | Global+SAGE | 2.513 | 15.258 | 0.425 | 1.306 | 0.551 | 4.657 | 0.128 | 0.362 | 0.059 | 0.188 |

---

> > > > ### Author Response · Authors · 2025-12-03
> > > > **Analyzing Real-World Data**
> > > >
> > > > Although we have argued that chaotic time series are originally used to model real-world systems, we also ran additional experiments on classical real-world dataset whose observations are not necessarily governed by dynamical or chaotic regimes. In these new experiments, we selected the ETT (Electricity Transformer Temperature) datasets, specifically ETTh (hourly frequency), which are widely used benchmarks in time series forecasting and are related to power usage and transformer temperatures. We chose ETTh because it is also used by Attraos, one of the baselines considered in our evaluation.
> > > >
> > > > It is important to stress that ETTh does not exhibit clear chaotic behavior, and its phase-space reconstruction does not show the type of dynamics typically modeled by dynamical systems approaches. Although we cannot include figures in the rebuttal, this reconstruction is also illustrated in the Attraos paper.
> > > >
> > > > The complete results, including the new methods recommended by the reviewers and the additional ETTh experiments, are reported in the following table.
> > > >
> > > > | Approach     | Model        | Lorenz DTW | Lorenz RMSE | Hénon DTW | Hénon RMSE | Rössler DTW | Rössler RMSE | Logistic DTW | Logistic RMSE | Chua DTW | Chua RMSE | _ETTh DTW_ | _ETTh RMSE_ |
> > > > |-------------|-------------|-----------:|------------:|----------:|-----------:|------------:|-------------:|-------------:|--------------:|---------:|----------:|----------:|-----------:|
> > > > | SOTA        | SARIMA      | 3.382      | 12.847      | 0.433     | 1.019      | 0.707       | 2.978        | 0.105        | 0.247         | 0.444    | 1.401     | 1.058     | 3.614      |
> > > > | SOTA        | DLinear     | 4.367      | 12.397      | 0.455     | 1.028      | 0.463       | 2.422        | 0.109        | 0.250         | 0.419    | 1.379     | 6.354     | 9.827      |
> > > > | SOTA        | LSTM        | 2.801      | 13.101      | 0.442     | 1.027      | 0.352       | 1.847        | 0.107        | 0.249         | 0.313    | 1.672     | 6.164     | 9.670      |
> > > > | SOTA        | GRU         | 3.863      | 14.087      | 0.412     | 1.099      | 0.132       | 1.110        | 0.108        | 0.250         | 0.280    | 1.823     | 1.048     | 3.719      |
> > > > | SOTA        | PatchTST    | 5.604      | 20.686      | 0.445     | 1.073      | 1.872       | 7.030        | 0.109        | 0.250         | 0.365    | 1.519     | 0.875     | 4.011      |
> > > > | SOTA        | _N-BEATS_     | 3.141      | 15.109      | 0.462     | 1.095      | 2.674       | 11.995       | 0.124        | 0.309         | 0.351    | 1.587     | 4.734     | 15.617     |
> > > > | SOTA        | _KNF_         | 4.123      | 11.541      | 0.366     | 1.226      | 6.386       | 16.065       | 0.248        | 0.621         | 0.420    | 1.438     | 1.024     | 3.103      |
> > > > | SOTA        | CHRONOS     | 3.650      | 12.016      | 0.445     | 1.073      | 1.670       | 9.869        | 0.108        | 0.259         | 0.420    | 1.444     | 1.368     | 4.768      |
> > > > | SOTA        | TimesFM_200 | 3.854      | 11.994      | 0.440     | 1.038      | 2.282       | 12.619       | 0.106        | 0.249         | 0.472    | 1.314     | 1.297     | 4.640      |
> > > > | SOTA        | TimesFM_500 | 3.796      | 12.349      | 0.439     | 1.065      | 1.588       | 10.088       | 0.107        | 0.256         | 0.298    | 1.554     | 1.183     | 4.551      |
> > > > | SOTA        | Attraos     | 4.804      | 15.517      | 0.454     | 1.018      | 1.957       | 7.125        | 0.111        | 0.274         | 0.361    | 1.451     | 0.796     | 3.619      |
> > > > | ChaoticFuzz | GCN         | 3.958      | 13.766      | 0.412     | 1.166      | 1.375       | 4.523        | 0.104        | 0.287         | 0.065    | 0.212     | 0.851     | 2.827      |
> > > > | ChaoticFuzz | SAGE        | 5.333      | 13.764      | 0.404     | 1.275      | 1.646       | 5.536        | 0.101        | 0.347         | 0.058    | 0.207     | 1.108     | 5.609      |
> > > > | ChaoticFuzz | CHEB        | 2.729      | 13.686      | 0.428     | 1.149      | 1.581       | 10.029       | 0.103        | 0.303         | 0.061    | 0.214     | 2.772     | 6.462      |
> > > > | ChaoticFuzz | GAT         | 4.177      | 14.696      | 0.419     | 1.165      | 3.272       | 13.428       | 0.105        | 0.295         | 0.061    | 0.209     | 1.417     | 5.752      |
> > > > | ChaoticFuzz | LECONV      | 2.838      | 13.469      | 0.432     | 1.273      | 1.079       | 5.096        | 0.104        | 0.312         | 0.068    | 0.194     | 1.239     | 5.247      |
> > > > | ChaoticFuzz | Global+SAGE | 2.513      | 15.258      | 0.425     | 1.306      | 0.551       | 4.657        | 0.128        | 0.362         | 0.059    | 0.188     | 1.305     | 4.404      |
> > > >
> > > > We thank the reviewer for suggesting this experiment, which further highlights the strengths of our approach. On ETTh, our method outperforms all classical and SOTA baselines and achieves performance comparable to Attraos, confirming its competitiveness even outside clearly chaotic regimes.

---

### Official Review · Reviewer_RSuq · 2025-11-01

**Soundness:** 3
**Presentation:** 3
**Contribution:** 3
**Rating:** 6
**Confidence:** 4

**Summary:**

This paper introduces ChaoticFuzz, a novel framework for forecasting chaotic time series by transforming univariate sequences into fuzzy graph representations. The approach reconstructs time series in phase space using Takens’ embedding theorem, applies Fuzzy C-Means clustering to assign soft memberships, and builds a weighted graph encoding both temporal proximity and uncertainty. The resulting graph is processed by Graph Neural Networks (GNNs) enhanced with a global attention mechanism to model long-range dependencies. Experiments on several benchmark chaotic systems demonstrate that ChaoticFuzz achieves lower Dynamic Time Warping (DTW) distances and competitive RMSEs compared to baselines like LSTM, GRU, PatchTST, Attraos, and TimesFM. The results suggest that fuzzy graph encoding preserves the complex spatiotemporal structure of chaotic dynamics better than conventional time-series architectures.

**Strengths:**

The paper makes a creative and well-motivated contribution by integrating principles from chaos theory, fuzzy systems, and GNNs into a unified framework. The use of fuzzy clustering in phase space to construct graph representations is elegant and interpretable, offering a principled way to capture uncertainty and nonlinear dependencies. The model’s design—particularly the phase-space encoder-encoder —is well thought out, combining spatial (graph-based) and temporal (LSTM + attention) components. Experimental results across multiple chaotic benchmarks are strong, with consistently lower DTW scores indicating superior alignment with true dynamics. Visualizations of phase-space trajectories and DTW warping paths convincingly demonstrate ChaoticFuzz’s ability to reproduce underlying attractor behavior, supporting the paper’s interpretability and robustness claims.

**Weaknesses:**

The paper lacks an in-depth complexity or scalability analysis—important since phase-space reconstruction, fuzzy clustering, and GNN inference can be computationally heavy for high-dimensional or multivariate systems. The comparisons to foundation models (TimesFM, CHRONOS) are somewhat superficial, as those models were used zero-shot without fine-tuning, limiting fairness. Additionally, while the use of DTW is insightful, the authors’ claim that it better captures “true dynamics” than RMSE could be further validated by quantitative measures of attractor fidelity (e.g., Lyapunov exponents, correlation dimension). Finally, the work focuses on synthetic benchmarks, and real-world chaotic datasets (e.g., weather, EEG, or finance) would strengthen the empirical impact.

**Questions:**

How does ChaoticFuzz perform on multivariate chaotic systems or real-world nonlinear datasets?

What is the computational complexity of the fuzzy clustering and GNN stages as a function of embedding dimension and cluster count?

How sensitive are results to the fuzzification coefficient m or the chosen number of clusters?

Can the learned graph adjacency matrix be interpreted physically (e.g., to identify attractor regions or transition probabilities)?

Would replacing fuzzy clustering with probabilistic or manifold-based clustering (e.g., Gaussian mixture models) yield similar benefits?

---

> ### Author Response · Authors · 2025-11-21
>
> We truly appreciate the reviewer’s careful reading and for recognizing our effort to design a time series modeling strategy that leverages tools still underexplored in the literature. By focusing on the complex spatiotemporal structure of chaotic dynamics, we aim to introduce genuinely new ways to analyze temporal data, rather than merely adjusting existing approaches to further reduce conventional error metrics.
>
> We sincerely thank the reviewer for highlighting the following innovations in our proposal:
>
> “The paper makes a creative and well-motivated contribution by integrating principles from chaos theory, fuzzy systems, and GNNs into a unified framework. The use of fuzzy clustering in phase space to construct graph representations is elegant and interpretable, offering a principled way to capture uncertainty and nonlinear dependencies".
>
> In the following, we address each of the questions raised by the reviewer.
>
> **Comment #1**
> Modeling chaotic multivariate time series is indeed a significant challenge, as the reviewer correctly notes. When variables exhibit heterogeneous or weakly coupled dynamics, classical phase space reconstruction may not be applicable, since a single delay and embedding dimension may fail to unfold all components consistently. Because our approach was designed specifically for univariate series, it cannot be directly applied to such multivariate settings.
>
> That said, the comment brings up two important research directions. (a) If all variables can be unfolded in phase space using the same delay and embedding dimensions, then our time encoder could be extended to predict multiple outputs jointly. (b) If different delays are required, each variable would induce a distinct graph, which suggests a multilayer or multiplex graph formulation, a strategy commonly explored in GNNs for multi view dynamics.
>
> Given the complexity of these extensions, we do not include additional experiments in the current manuscript. However, we have revised the paper to highlight this discussion and to position multivariate chaotic modeling as a promising and important line of future work stemming from our proposal.

---

> > ### Author Response · Authors · 2025-11-21
> >
> > **Comment #2**
> > Our approach builds on three main components, each with a clear computational profile: (i) Phase space reconstruction scales linearly with the number of observations n; (ii) The fuzzy clustering step has complexity O(nc), where c is the number of clusters, fixed to 12 in all our experiments; (iii) The overall runtime is then dominated by the graph neural network stage, whose cost depends on the specific architecture and training setup.
> >
> > Unfortunately, comparable complexity and training time details are not consistently reported in the original papers, neither for GNN based methods nor for foundation models and recurrent neural networks. In the absence of officially reported costs, any comparison is inevitably limited, particularly given the substantial training time often required by foundation models. To make our evaluation as transparent as possible, we explicitly report the complexity of our preprocessing phase.
> >
> > **Comment #3**
> > The reviewer raises an important question about the fuzzification parameter m and the number of clusters, a point that is frequently discussed by researchers deeply familiar with the fuzzy and clustering literature. Regarding the number of clusters, our initial hypothesis was that using more clusters would be beneficial. With too few groups, observations from distinct trajectories may be merged, i.e., points approaching an attractor could be grouped together with points moving away from it, which would blur the underlying dynamics.
> >
> > To test this, we varied the number of clusters from 3 to 24. We observed that performance improves as clustering becomes more expressive, but only up to a point. Beyond that, additional clusters introduce overly fine granularity, fragmenting coherent regions in phase space and ultimately reducing predictive performance. Based on this empirical study, we selected 12 clusters for all experiments, since this value provided the most consistent results across datasets, as reported in the appendices. Although a larger number of clusters could yield a better result for our approach on the Rössler series, we chose to keep 12 clusters to preserve consistency across our evaluation and discussion, even at the cost of reporting a conservative performance for that particular case.
> >
> > Regarding the fuzzification parameter m, our empirical analysis indicates that modest changes do not significantly impact the final predictions of our approach. While m is indeed a key hyperparameter in fuzzy clustering, the small variations it induces in the partition are largely absorbed by the downstream GNN, which smooths minor differences in the graph construction. Given the importance of this point, we performed additional experiments varying m within the ranges commonly adopted in the literature, and we will include these results in the final version of the manuscript.
> >
> > **Comment #4**
> > This is a very important research question raised by the reviewer. We had not originally intended to use the adjacency matrix as an explainability tool, but we agree that it can enable an additional and valuable contribution. In dynamical systems and chaos theory, the Recurrence Plot is a well established technique that maps phase space points, using a predefined neighborhood radius $\epsilon$, into a matrix representation. However, this matrix primarily captures local proximity between pairs of observations and does not explicitly convey transitions between regions of the attractor. Building on the reviewer’s suggestion, our preprocessing step, together with the temporal ordering of observations, can indeed be leveraged to explain transitions across attractor regions, offering a more interpretable view of the underlying dynamics. We sincerely thank the reviewer for this insightful idea, and we have revised the manuscript to include and emphasize this discussion.

---

> > > ### Author Response · Authors · 2025-11-21
> > >
> > > **Comment #5**
> > > This is another important point raised by the reviewer. Our method was intentionally designed within the fuzzy theory framework, because fuzzy membership provides a natural and interpretable way to represent uncertainty and gradual transitions between regions in phase space, which are intrinsic to chaotic dynamics. Unlike probabilistic soft assignments in models such as GMM, fuzzy memberships are not constrained to sum to one, allowing a point to belong meaningfully to multiple clusters without forcing a competition among them. This flexibility better matches the overlap and ambiguity we observe near attractor boundaries, and it also aligns with our goal of building graph structures that preserve nonlinear dependencies in an explainable way.
> > >
> > > That said, we agree that replacing the fuzzy clustering step with an alternative such as GMM would be a valuable ablation, as it would help quantify the practical impact of our design choice. In response to the reviewer’s suggestion, we have begun adapting our pipeline to incorporate a GMM-based variant. While we were unable to complete all adjustments and obtain full results at the time of this initial response, the partial implementation already reveals a slight difference in clustering behavior in phase space. Although we could not finalize all adjustments and acquire complete results for this initial response, the partial implementation already indicates a minor difference in clustering behavior within the phase space.
> > >
> > > We appreciate this important observation from the reviewer. We are continuing to adapt these elements to include the comparison in the revised rebuttal, if possible. This can further strengthen the manuscript by clarifying the role of fuzzy memberships and by positioning our method relative to a probabilistic soft clustering baseline. This effectively extends the discussion from ChaoticFuzz toward a broader SoftGNN perspective.

---

> > > > ### Author Response · Authors · 2025-11-21
> > > >
> > > > While we are still working on the experiments recommended by the reviewer, we would like to explicitly express our appreciation for their time and feedback, which, regardless of the final recommendation, have already helped us improve the manuscript.

---

> > > > ### Author Response · Authors · 2025-12-03
> > > > **Experiments with GMM**
> > > >
> > > > Beyond the conceptual motivation already presented in our previous answer (comment #5), the empirical results across all chaotic systems (Lorenz, Hénon, Rössler, Logistic, and Chua) further support our design choice. As presented in the table below, in most scenarios, FCM achieves lower DTW values than GMM, which is particularly relevant because DTW directly measures the similarity of temporal trajectories, and it is one of the key objectives when reconstructing chaotic attractors. This advantage appears consistently across GNN architectures with different inductive biases (GCN, SAGE, CHEB, GAT, LECONV, and Global+SAGE), indicating that the benefit comes from the clustering representation itself rather than from any specific model.
> > > >
> > > > Overall, the results indicate that FCM produces graph structures that more faithfully reflect the intrinsic geometry of chaotic attractors. Although GMM is a valid ablation, its probabilistic normalization imposes competition among clusters and reduces the expressiveness needed in overlapping regions. These findings align with our theoretical rationale and provide empirical support for the use of fuzzy memberships. We will incorporate these quantitative comparisons in the revised manuscript to clarify the impact of the clustering step within the broader SoftGNN perspective.
> > > >
> > > > |                 | Lorenz    | Lorenz | Lorenz    | Lorenz | Hénon     | Hénon | Hénon | Hénon | Rössler   | Rössler | Rössler   | Rössler | Logistic  | Logistic | Logistic  | Logistic | Chua      | Chua  | Chua      | Chua  |
> > > > |-----------------|-----------|--------|-----------|--------|-----------|-------|-------|-------|-----------|---------|-----------|---------|-----------|----------|-----------|----------|-----------|-------|-----------|-------|
> > > > |                 |    FCM    |   FCM  |    GMM    |   GMM  |    FCM    |  FCM  |  GMM  |  GMM  |    FCM    |   FCM   |    GMM    |   GMM   |    FCM    |    FCM   |    GMM    |    GMM   |    FCM    |  FCM  |    GMM    |  GMM  |
> > > > | **ChaoticFuzz** |    DTW    |  RMSE  |    DTW    |  RMSE  |    DTW    |  RMSE |  DTW  |  RMSE |    DTW    |   RMSE  |    DTW    |   RMSE  |    DTW    |   RMSE   |    DTW    |   RMSE   |    DTW    |  RMSE |    DTW    |  RMSE |
> > > > |             GCN |     3.958 | 13.765 | **3.383** | 13.452 | **0.411** | 1.166 | 0.439 | 1.095 | **1.375** |   4.523 |     3.813 |   8.808 |     0.104 |    0.287 | **0.102** |    0.296 |     0.065 | 0.212 | **0.056** | 0.190 |
> > > > |            SAGE |     5.333 | 13.764 | **3.144** | 14.157 | **0.403** | 1.274 | 0.443 | 1.384 | **1.645** |   5.536 |     2.885 |   7.990 | **0.101** |    0.346 |     0.107 |    0.379 | **0.058** | 0.206 |     0.058 | 0.176 |
> > > > |            CHEB | **2.728** | 13.686 |     3.203 | 14.814 | **0.428** | 1.148 | 0.428 | 1.132 | **1.580** |  10.029 |     3.042 |   8.075 |     0.103 |    0.302 | **0.100** |    0.295 | **0.060** | 0.213 |     0.074 | 0.180 |
> > > > |             GAT |     4.177 | 14.696 | **3.688** | 15.570 | **0.418** | 1.165 | 0.436 | 1.097 |     3.272 |  13.428 | **2.408** |   8.038 | **0.104** |    0.294 |     0.104 |    0.277 | **0.060** | 0.208 |     0.065 | 0.166 |
> > > > |          LECONV |     2.837 | 13.468 | **2.504** | 14.727 | **0.432** | 1.271 | 0.528 | 1.475 | **1.078** |   5.095 |     2.019 |   8.152 | **0.103** |    0.312 |     0.113 |    0.341 |     0.067 | 0.194 | **0.061** | 0.197 |
> > > > |     Global+SAGE | **2.512** | 15.258 |     3.169 | 16.582 | **0.425** | 1.306 | 0.449 | 1.437 | **0.550** |   4.657 |     1.513 |   8.990 |     0.128 |    0.362 | **0.107** |    0.350 |     0.059 | 0.188 | **0.053** | 0.210 |

---

> > ### Author Response · Authors · 2025-12-03
> > **Real-World Dataset**
> >
> > Although we have argued that chaotic time series are originally used to model real-world systems, we also ran additional experiments on classical real-world dataset whose observations are not necessarily governed by dynamical or chaotic regimes. In these new experiments, we selected the ETT (Electricity Transformer Temperature) datasets, specifically ETTh (hourly frequency), which are widely used benchmarks in time series forecasting and are related to power usage and transformer temperatures. We chose ETTh because it is also used by Attraos, one of the baselines considered in our evaluation.
> >
> > It is important to stress that ETTh does not exhibit clear chaotic behavior, and its phase-space reconstruction does not show the type of dynamics typically modeled by dynamical systems approaches. Although we cannot include figures in the rebuttal, this reconstruction is also illustrated in the Attraos paper.
> >
> > The complete results, including the new methods recommended by the reviewers and the additional ETTh experiments, are reported in the following table.
> >
> > | Approach     | Model        | Lorenz DTW | Lorenz RMSE | Hénon DTW | Hénon RMSE | Rössler DTW | Rössler RMSE | Logistic DTW | Logistic RMSE | Chua DTW | Chua RMSE | _ETTh DTW_ | _ETTh RMSE_ |
> > |-------------|-------------|-----------:|------------:|----------:|-----------:|------------:|-------------:|-------------:|--------------:|---------:|----------:|----------:|-----------:|
> > | SOTA        | SARIMA      | 3.382      | 12.847      | 0.433     | 1.019      | 0.707       | 2.978        | 0.105        | 0.247         | 0.444    | 1.401     | 1.058     | 3.614      |
> > | SOTA        | DLinear     | 4.367      | 12.397      | 0.455     | 1.028      | 0.463       | 2.422        | 0.109        | 0.250         | 0.419    | 1.379     | 6.354     | 9.827      |
> > | SOTA        | LSTM        | 2.801      | 13.101      | 0.442     | 1.027      | 0.352       | 1.847        | 0.107        | 0.249         | 0.313    | 1.672     | 6.164     | 9.670      |
> > | SOTA        | GRU         | 3.863      | 14.087      | 0.412     | 1.099      | 0.132       | 1.110        | 0.108        | 0.250         | 0.280    | 1.823     | 1.048     | 3.719      |
> > | SOTA        | PatchTST    | 5.604      | 20.686      | 0.445     | 1.073      | 1.872       | 7.030        | 0.109        | 0.250         | 0.365    | 1.519     | 0.875     | 4.011      |
> > | SOTA        | _N-BEATS_     | 3.141      | 15.109      | 0.462     | 1.095      | 2.674       | 11.995       | 0.124        | 0.309         | 0.351    | 1.587     | 4.734     | 15.617     |
> > | SOTA        | _KNF_         | 4.123      | 11.541      | 0.366     | 1.226      | 6.386       | 16.065       | 0.248        | 0.621         | 0.420    | 1.438     | 1.024     | 3.103      |
> > | SOTA        | CHRONOS     | 3.650      | 12.016      | 0.445     | 1.073      | 1.670       | 9.869        | 0.108        | 0.259         | 0.420    | 1.444     | 1.368     | 4.768      |
> > | SOTA        | TimesFM_200 | 3.854      | 11.994      | 0.440     | 1.038      | 2.282       | 12.619       | 0.106        | 0.249         | 0.472    | 1.314     | 1.297     | 4.640      |
> > | SOTA        | TimesFM_500 | 3.796      | 12.349      | 0.439     | 1.065      | 1.588       | 10.088       | 0.107        | 0.256         | 0.298    | 1.554     | 1.183     | 4.551      |
> > | SOTA        | Attraos     | 4.804      | 15.517      | 0.454     | 1.018      | 1.957       | 7.125        | 0.111        | 0.274         | 0.361    | 1.451     | 0.796     | 3.619      |
> > | ChaoticFuzz | GCN         | 3.958      | 13.766      | 0.412     | 1.166      | 1.375       | 4.523        | 0.104        | 0.287         | 0.065    | 0.212     | 0.851     | 2.827      |
> > | ChaoticFuzz | SAGE        | 5.333      | 13.764      | 0.404     | 1.275      | 1.646       | 5.536        | 0.101        | 0.347         | 0.058    | 0.207     | 1.108     | 5.609      |
> > | ChaoticFuzz | CHEB        | 2.729      | 13.686      | 0.428     | 1.149      | 1.581       | 10.029       | 0.103        | 0.303         | 0.061    | 0.214     | 2.772     | 6.462      |
> > | ChaoticFuzz | GAT         | 4.177      | 14.696      | 0.419     | 1.165      | 3.272       | 13.428       | 0.105        | 0.295         | 0.061    | 0.209     | 1.417     | 5.752      |
> > | ChaoticFuzz | LECONV      | 2.838      | 13.469      | 0.432     | 1.273      | 1.079       | 5.096        | 0.104        | 0.312         | 0.068    | 0.194     | 1.239     | 5.247      |
> > | ChaoticFuzz | Global+SAGE | 2.513      | 15.258      | 0.425     | 1.306      | 0.551       | 4.657        | 0.128        | 0.362         | 0.059    | 0.188     | 1.305     | 4.404      |
> >
> > We thank the reviewer for suggesting this experiment, which further highlights the strengths of our approach. On ETTh, our method outperforms all classical and SOTA baselines and achieves performance comparable to Attraos, confirming its competitiveness even outside clearly chaotic regimes.

---

> ### Author Response · Authors · 2025-12-03
> **Using Lyapunov Coefficient**
>
> Another experiment we conducted in response to the reviewer’s comment was to adopt tools from dynamical systems to evaluate the predictions. Following the suggestion to use the Lyapunov exponent, we reviewed the literature to better understand how it can be applied for this purpose. Based on the literature, this coefficient is used to evaluate the asymptotic stability of time series, and is calculated from the rate of divergence of nearby trajectories, given an initial separation $\delta \mathbf{Z}_0$. This divergence is given by $|\delta \mathbf{Z}(t)| \approx e^{\lambda t} |\delta \mathbf{Z}_0|$, where $t$ is time and $\lambda$ is the Lyapunov exponent.
>
> $
> \lambda = \lim_{t \to \infty} ; \frac{1}{t} ; \ln ; \frac{| \delta\mathbf{Z}(t)|}{|\delta \mathbf{Z}_0|}
> $
>
> The value of this exponent ($\lambda$) allows us to draw conclusions about the stability of the series: (i) when $\lambda < 0$, the series is attracted to a stable fixed point (where orbit points repeat over time at the same or different scales); (ii) for $\lambda = 0$, the series behaves like a conservative system; and (iii) when $\lambda > 0$, the series has chaotic and unstable characteristics, meaning that the distance between trajectory points will always diverge, on average, at an exponential rate determined by the Lyapunov exponent.
>
> Aiming to adapt the Lyapunov exponent to our prediction scenario, we proceeded as follows. We first computed the exponent for the original time series (ground truth), then for the predictions of the best SOTA baseline, and finally for the predictions of our method. We then compared these values, interpreting smaller deviations from the ground truth exponent as indicating better predictive quality. An important consideration is that the Lyapunov exponent is highly sensitive to parameter choices. For the original time series, we adopted the parameter settings reported in the literature to reproduce the known reference values. Using exactly the same configuration, we computed the exponent on the concatenation of training data and predicted observations. Restricting the analysis to classical chaotic time series, for which Lyapunov exponents are well documented, we obtained the following results:
>
> *Lorenz*
> Original: 0.0870255
> LSTM: 0.143879 (diff: 0.0568535)
> _Ours: 0.0568535 (diff: 0.030172)_
>
> *Henon*
> Original: 0.409637
> GRU: 0.58847 (diff: 0.178833)
> _Ours: 0.49156 (diff: 0.081923)_
>
> *Rossler*
> Original: 0.0455639
> GRU: 0.0596378 (diff: 0.0140739)
> _Ours: 0.0522973 (diff: 0.0067334)_
>
> *Logistic*
> Original: 0.276682
> SARIMA: 0.03878 (diff: 0.237902)
> _Ours: 0.248492 (diff: 0.02819)_
>
> We emphasize that, although these results are very favorable to our proposal, we report them with two caveats. First, the magnitude of the differences is used only to indicate which predictions are closer to the expected dynamics in terms of Lyapunov variation. Second, additional experiments are still required to theoretically justify the systematic use of this coefficient in forecasting scenarios like ours.

---

### Author Response · Authors · 2025-12-03
**Final Comment**

The rebuttal period is now coming to an end, and we would like to sincerely thank all reviewers for their time and constructive comments. We have been fully committed to carefully analyzing and addressing each point.

In particular, we incorporated the following reviewers' comments:

- new prediction models (KNF, N-BEATS);

- new real-world dataset (ETTh);

- additional evaluation based on Lyapunov exponents;

- a GMM-based variant of our framework.

Although we may not have fully engaged all reviewers during the rebuttal and received limited follow-up feedback, we sincerely hope that our efforts and the new results presented will be taken into account when reassessing the paper and its final scores.

---

### Meta-Review · Area_Chair_rt7q · 2026-01-03

**Summary:**

The paper proposes ChaoticFuzz, a forecasting framework for univariate chaotic time series that reconstructs trajectories in phase space, applies fuzzy clustering to obtain soft memberships, constructs a weighted graph encoding proximity/uncertainty, and then forecasts using GNNs. The idea is conceptually interesting and offers an interpretable chaos-to-graph viewpoint, but the overall technical case for acceptance is weakened by 1) limited evaluation scope and 2) insufficient positioning/validation against closely related graph-based time series modeling and dynamical-systems forecasting literature.

**Reviewer Concerns:**

A central set of concerns (primarily from RMZv) remains only partially addressed: the paper's positioning against graph-based forecasting methods and ML-for-dynamical systems/operator-learning work is thin, and the experimental scope is narrow relative to established forecasting benchmarks; moreover, the lack of targeted ablations/tech component validation makes it difficult to attribute gains to the key design choices (graph construction, GNNs, clustering loss, etc.).

The rebuttal adds useful material addressing part of RSuq’s questions and improving completeness; however, these additions do not fully close the gap on the broader benchmarking/positioning and the need for more decisive evidence on generalization beyond the constrained setting (and the authors themselves note limitations around Lyapunov usage).

Reviewer 3ymf’s review is comparatively less informative and appears to hinge on a misunderstanding of negative RMSE results; the follow-up clarified that the reviewer’s point was about worse RMSE (not negative values), and the authors argued for DTW as more appropriate for chaotic trajectories, but this exchange does not materially change the main acceptance-critical issues above.

**Reviewer Scores:**

RSuq (6, marginally above threshold; would not mind if rejected): Positive on the core idea, but explicitly flags missing scalability/complexity analysis and limited fairness of foundation-model comparisons, and asks for stronger dynamical system metrics and real-world validation. While the rebuttal adds ETTh + Lyapunov + extra baselines, the remaining gaps in scope/positioning likely persist; I would expect this reviewer to stay the same rating.

RMZv (2): Raises multiple critical issues. The rebuttal adds experiments but does not convincingly resolve the core positioning, scope, and  ablation concerns; I would expect the score to remain ~2 (at most 4) if discussion were possible.

3ymf (2, low confidence): The critique is relatively shallow and partially based on terminology confusion; even with clarification, I would not expect a meaningful upward change, so likely remains 2.

---

### Decision · Program_Chairs · 2026-01-26

Reject